# The defensome of prokaryotes in aquifers

Pengwei Li [1,2], Zongzhi Wu [1,3], Tang Liu[4], Chunfang Deng[5], Quan Liu[6] & Jinren Ni [1,5] ✉

Groundwater harbors a pristine biosphere where microbes co-evolve with less human interference, yet the ancient and ongoing arms race between prokaryotes and viruses remains largely unknown in such ecosystems. Based on our recent nationwide groundwater monitoring campaign across China, we construct a metagenomic groundwater prokaryotic defensome catalogue (GPDC), encompassing 190,810 defense genes, 90,824 defense systems, 139 defense families, and 669 defense islands from 141 prokaryotic phyla. Over 94% of the defense genes in GPDC are novel and contribute vast microbial immune resources in groundwater. We find that candidate phyla radiation (CPR) bacteria possess higher defense system density and diversity against intense phage infection, while microbes as a whole exhibit an inverse relationship between defense systems and adaptive traits like resistance genes in groundwater. We further identify five first-line defense families covering 69.2% of the total defense systems, and high-turnover accessory immune genes are mostly conveyed to defense islands by mobile genetic elements. Our study also reveals viral resistance to microbial defense through co-localized anti-defense genes and interactions between CRISPR-Cas9 and anti-CRISPR protein. These findings expand our understanding of microbial immunity in pristine ecosystems and offer valuable immune resources for potential biotechnological applications.

Prokaryotes have co-evolved with their viruses (phages) for nearly 400 million years[1,2]. Phages are extremely abundant in many ecosystems[3], with an estimate of approximately $10^{31}$ virus particles in the biosphere[4]. Phage infection is a major driver of prokaryotic evolution, accounting for about 20% of daily bacterial mortality, with more than $10^{23}$ infections occurring per second in the oceans alone[5]. To combat the constant predation pressure, prokaryotes have developed a variety of defense systems[6] capable of regulating the flux of genetic information spread by mobile genetic elements (MGEs) via horizontal gene transfer (HGT)[7,8], which are defined as single genes or groups of genes that confer partial or full resistance against phages[1]. These defense systems

are classified into two major groups (i.e., innate and adaptive immune systems) based on their components and function mechanisms[9,10]. For instance, the restriction-modification (R-M) system, as a typical innate immunity, utilizes DNA methylation to identify and cleave invading phage DNA[11]. By contrast, the CRISPR-Cas (clustered regularly interspaced short palindromic repeats and CRISPR-associated protein)[12] system enables adaptive immunity by acquiring and incorporating foreign phage sequences as host spacers. Ten novel anti-MGE systems were identified utilizing the preferred colocalization of defense systems in defense islands[13,14], marking a rapid expansion in the discovery of bacterial defense systems[15]. To date, over 150 defense systems have

[1]Environmental Microbiome and Innovative Genomics Laboratory, College of Environmental Sciences and Engineering, Peking University, Beijing 100871, P. R. China. [2]College of Environmental Sciences and Engineering, Key Laboratory of Water and Sediment Sciences, Ministry of Education, Peking University, Beijing 100871, P. R. China. [3]Environmental Protection Key Laboratory of All Materials Fluxes in River Ecosystems, Ministry of Ecology and Environment, Beijing 100871, P. R. China. [4]Environmental Microbiome Engineering and Innovative Genomics Laboratory, College of Chemistry and Environmental Engineering, Shenzhen University, Shenzhen 518060, P. R. China. [5]Eco-environment and Resource Efficiency Research Laboratory, School of Environment and Energy, Peking University Shenzhen Graduate School, Shenzhen 518055, P. R. China. [6]Peking-Tsinghua Center for Life Sciences, State Key Laboratory of Gene Function and Modulation Research, Peking-Tsinghua-NIBS Graduate Program, School of Life Sciences, Peking University, Beijing 100871, P. R. China. ✉e-mail: jinrenni@pku.edu.cn

been described, revealing a diversity of molecular mechanisms far greater than previously understood[16]. Consequently, the prokaryotic immune system is now recognized as much more complex than previously envisaged (confined to R-M, CRISPR-Cas, and a few other systems), and the pace of new discoveries shows no signs of slowing[1,8,13,16,17].

The ongoing arms race between prokaryotes and phages drives the evolution of diverse defense systems, making it essential to identify which systems are present within a genome to understand phage-host interactions[18]. Although the abundance and distribution of defense systems have been extensively studied in a few culturable model microorganisms[19–21], little is known about these defense systems in the vast number of unculturable microorganisms in natural environments[8,22]. Recent advances in bioinformatics and the establishment of microbial defense datasets[18,23] enable identifying uncultured microbial anti-MGE systems[24]. Beavogui, A. et al.[22] first proposed the term "defensome" and reported that bacterial communities exhibit diverse defense strategies against phages across a wide range of uncultured microbes in soil, marine, and human gut systems[22].

Groundwater is a critical freshwater resource, with microbes as the dominant living organisms[25,26] playing a crucial role in global biogeochemical cycling processes[27,28]. Previous studies on microbial communities have been limited to small-scale[29] or geologically specific areas[30], leaving a significant gap in continental or global-scaled studies in groundwater ecosystems[22,24]. Candidate phyla radiation (CPR) bacteria and DPANN (an acronym formed from the initials of the first five lineages discovered: Diapherotrites, Parvarchaeota, Aenigmarchaeota, Nanohaloarchaeota, and Nanoarchaeota) archaea represent ultrasmall symbiotic microorganisms[31–33] mostly detected in oxygen-limited environments, particularly in groundwater ecosystems[34,35]. Genome analyses[35,36] and experimental studies[31,32] reveal their distinctive features: reduced genome sizes and notable gaps in core metabolic potential, consistent with their symbiotic lifestyle. Understanding these microorganisms is important because their interactions with other microorganisms are likely to shape natural microbiome functions[32]. The intense phage infection targeting these abundant and widespread symbionts in groundwater[37] may lead to the development of a unique defensome, which requires further investigation. Given the typically anoxic or anaerobic environments and the limited human interference, groundwater presents an ideal pristine ecosystem for investigating microbial defensome as well as virus-host interactions and coevolution.

To unveil the unknown prokaryotic defensome in groundwater, we implemented a nationwide monitoring campaign across China and constructed a metagenomic groundwater prokaryotic defensome catalog (GPDC) from 29,031 metagenome-assembled genomes (MAGs). By identifying microbes for higher-density defense systems, major defense families for defense systems, defense islands for high-turnover accessory immune genes, and corresponding mobile genetic elements, we reveal the unique characteristics of prokaryotic defensome as well as the trade-offs between defense systems and adaptive traits (e.g., antibiotic resistance genes, heavy-metal resistance genes, and virulence factors) in groundwater. Interestingly, colocalized anti-defense genes and interactions between CRISPR-Cas9 and anti-CRISPR protein are also unraveled to help understand the viral resistance to microbial defense. Our study helps to unveil the interactions between prokaryotes and their phages in groundwater and enables us to perceive plentiful unknown defense mechanisms in the huge repositories of microbes in the subterranean world.

## Results and discussion
### The vast reservoir of prokaryotic defensome in groundwater
We established a metagenomic groundwater prokaryotic defensome catalog (GPDC) (see Supplementary Figs. S1, S2 for an overview), encompassing 190,810 defense genes, 90,824 defense systems, 139

defense families, and 669 defense islands derived from 27,578 bacterial and 1453 archaeal MAGs (≥ 70% completeness and ≤ 10% contamination; Supplementary Data 1 and 2). Notably, the majority (94.1%) of bacterial defense genes in GPDC ($n = 185,355$; Fig. 1B; see "Methods") are novel compared to those of the NCBI RefSeq complete prokaryotic database (Supplementary Data 3), indicating a vast untapped reservoir of microbial immune systems within groundwater. Bacterial defense genes are assigned to 88,166 defense systems that grouped into 139 defense families, in which R-M is the most prevalent defense family (found in about 50% of bacterial MAGs), followed by SoFIC, CRISPR-Cas, AbiE, and MazEF (present in > 10% of bacterial MAGs) (Fig. 1A). The analyses of bacterial defense genes (prevalence of type IIG R-M, type II R-M MTases, and type II R-M REases) further support the predominance of R-M systems (see Supplementary Fig. S3 for the distribution of defense genes), which is similar to previous findings of soil, human gut, and marine environments[22,38]. The five predominant defense families (R-M, SoFIC, CRISPR-Cas, AbiE, and MazEF) are defined as "first-line" defense here[39], collectively accounting for 69.2% of all detected bacterial defense systems, but have much lower diversity (3.6% of 139 defense families). These first-line systems protect microbes against viral attacks, in which R-M and CRISPR-Cas are also reported in studies at other environments[16,22]. In contrast, the remaining defense families (e.g., CBASS, Wadjet, and Septu) labeled as "accessory defense" are less abundant[6], representing 30.8% of the 88,166 defense systems, but exhibit much higher diversity, encompassing 96.4% of the 139 defense families. These accessory systems play complementary and non-negligible roles, forming a flexible accessory immune reservoir in phage-rich environments[6]. Both types of defense systems constitute the integral defensome with a broader range of phage defense[6,40]. In addition, 5455 defense genes and 2658 defense systems across 81 defense families are identified from 1453 archaeal genomes, in which R-M, AbiE, and CRISPR-Cas are also identified as first-line defense systems (Supplementary Fig. S4A, see Supplementary Fig. S4 for archaeal defensome) as in bacterial defense.

Among all bacterial MAGs, the number of defense systems per genome varies significantly, ranging from 0 to 83 (Fig. 1D). Most bacterial MAGs harbor limited defense systems, with an average of 3.2 per genome. Notably, 28% (7731) of the MAGs entirely lack defense systems, while a small subset (0.97%; 268 MAGs) harbor more than 20 defense systems (Fig. 1D). These "defense supercarriers" are mainly annotated as Pseudomonadota, possessing larger genome size, longer average gene lengths, and more coding sequences than other bacteria (Supplementary Fig. S5A–D). Moreover, they exhibit distinct physiological traits like lower optimal growth temperatures (Supplementary Fig. S5E). Interestingly, the distribution of defense systems follows an exponential decline (Fig. 1D), similar to patterns observed in marine bacteria[22]. This suggests that groundwater and marine ecosystems may also share other key features, such as low HGT rates[41] and comparable virus-to-prokaryote ratios (VPRs)[3], both of which could shape the defensome of these environments. In addition, when comparing the number of defense systems across ecosystems[22], MAGs in groundwater exhibit a higher average number of defense systems, suggesting more frequent virus-host interactions in groundwater (Fig. 1E; see "Methods"). Alternatively, DefensePredictor, a recently developed tool that utilizes protein language models to identify defense genes[42], is used on our 1626 high-quality MAGs (DefensePredictor results added in Supplementary Fig. S6). We find more candidate defensive genes (2.4 times than DefenseFinder, Supplementary Fig. S6A), which further highlights the huge potential of digging new defense genes in groundwater.

A defense system usually consists of two main components: a sensor that detects viral infection and an effector that either attacks the phage or eliminates the infected cell to prevent phage replication[1]. Effectors degrading viral nucleic acids account for 56.4% of all defense systems (average 1.8 per MAG; Fig. 1F). These effectors include both

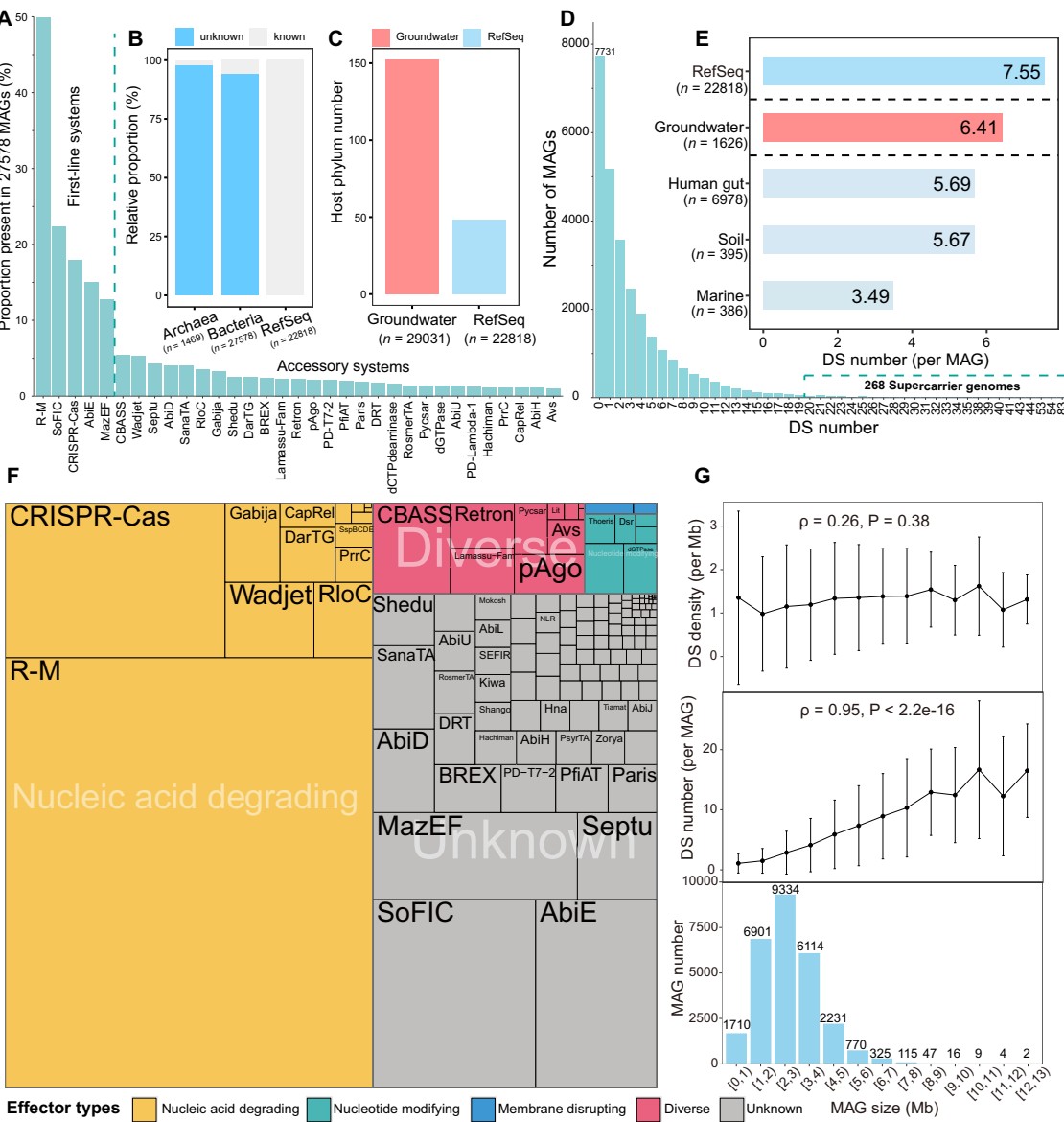

**Fig. 1 | The defensome of groundwater bacteria. A** Prevalence of defense families in groundwater bacterial MAGs, showing only families present in >1% of all MAGs. **B** Sequence novelty of groundwater defense genes compared to NCBI RefSeq prokaryotic genomes. The number of MAGs analyzed are shown as *n* values. **C** Prokaryotic host phyla diversity of defense genes in groundwater MAGs versus the RefSeq database (based on NCBI classification annotations). **D** Distribution of defense system (DS) number per MAG. **E** Average number of defense systems per MAG across datasets: 22,818 NCBI RefSeq prokaryotic genomes, high-quality genomes from human gut, soil, sea (from a previous study[22]), and 1626 high-quality

groundwater MAGs. **F** Relative abundance of defense systems in bacterial genomes. Treemap rectangles represent defense systems, with area proportional to mean abundance and color indicating effector type. **G** Variation in defense system number (per MAG), density (per MAG and per Mb), and MAG size (Mb) across 27,578 MAGs. Error bars represent standard deviations of the mean; correlation assessed by two-sided Spearman's rank test with exact P value provided. The number of MAGs analyzed is shown in the bar graph. Source data are provided as a Source Data file.

targeted (e.g., CRISPR-Cas, R-M systems) and untargeted mechanisms (e.g., CBASS, Gabija, and toxin-antitoxin systems)[43–45]. Targeted systems allow the host to survive phage infections by cleaving viral DNA or RNA, while untargeted systems often lead to cell death or dormancy, potentially preventing phage replication but also permitting recovery if the phage is neutralized[1,10,43–46]. Moreover, a substantial proportion (36.9%) of the immune arsenals consist of unknown effector-type systems (Fig. 1F and Supplementary Data 4), which may contain novel immune mechanisms against viral infections and warrant further investigation in the laboratory.

Consistent with findings based on NCBI RefSeq complete prokaryotic genomes[18], we observed a significant positive correlation ($\rho = 0.95$, $P < 2.2e\text{-}16$) between the number of defense

systems and genome size, but no significant correlation ($\rho = 0.26$, $P = 0.3835$) between defense system density and genome size (Fig. 1G). A stepwise regression analysis further supports such positive relationship between MAG size ($P < 2e\text{-}16$) or N50 ($P < 2e\text{-}16$) and the number of defense genes or systems in genomes (Supplementary Data 5). Our results (Supplementary Fig. S7A) also show a significant positive correlation ($\rho = 0.88$, $P = 0.0007$) between genome size and prophage encounters. These patterns likely reflect the greater need for defense systems in bacteria with larger genomes, which tend to experience higher levels of invasion by exogenous genetic materials[47–49]. In this regard, the larger genome size of "defense supercarriers" implies their increased need for defense.

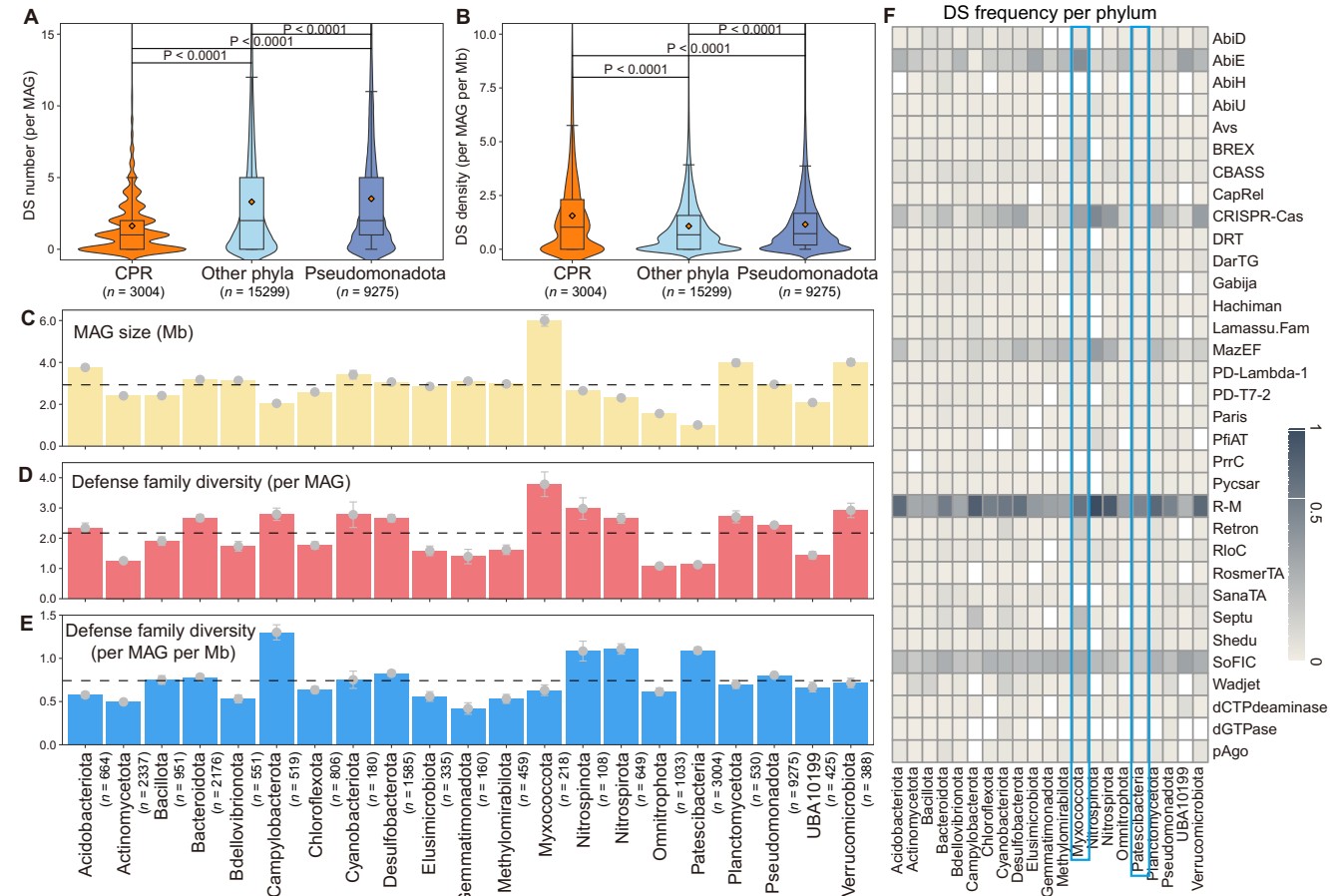

**Fig. 2 | Host characteristics of defense systems. A**, **B** Comparison of defense system (DS) number (per MAG) and density (per MAG and per Mb) among CPR, Pseudomonadota, and other phyla. Boxplots represent the 25th to 75th percentiles, the inner black line marks the median, whiskers extend to 1.5x the interquartile range, and the orange square dot represents the mean. Significant differences were assessed using the two-sided Wilcox test with Bonferroni-adjusted *P*-values. The number of MAGs analyzed are shown as *n* values. **C**–**E** MAG size, defense family diversity (per MAG), and defense family diversity density (per MAG and per Mb) across 21 dominant phyla (>100 MAGs). Error bars represent the 95% confidence interval (CI) of the mean; the horizontal dashed line represents the mean across all 21 phyla. **F** The heatmap shows the presence frequency of each defense system (rows) in each phylum (columns), with colors indicating frequency. Only defense families present in at least 1% of all MAGs are included for clarity. Source data are provided as a Source Data file.

Metagenomic analysis reveals slightly higher defense densities in reconstructed wells than in newly constructed wells (Supplementary Fig. S8C). We further compared the defense density across seven geographic zones (Supplementary Fig. S8E), and found the highest in zones I (Northeast Plain-Mountain) and VII (Qinghai-Tibet plateau Alpine frozen soil), and the lowest in zones III (South China bedrock low mountain foothill) and VI (Northwest arid desert), largely attributed to the distinct geo-environmental conditions. While the underlying mechanisms remain unclear, these findings suggest the complex effects of ecological contexts on the composition of microbial community and their defensome.

Some defense genes might have multiple complex functions beyond mitigating viral infection, while the DefenseFinder[18], a tool employing predefined HMM profiles and rules (a list of mandatory, accessory, or forbidden proteins necessary for the detection of a given system, along with the corresponding genetic architecture), could help to filter results and ensure low bias. Further, to partially address the limitations due to the lack of multi-omics data from the 607 groundwater samples, we conducted genetic experiments to validate the functions of the representative defense systems against phage infection as predicted (see Supplementary Fig. S9 for plaque assays). Future studies incorporating multi-omics approaches and functional experiments are essential to further validate and expand upon these findings.

## Defensome characteristics of different microbial lineages

Aquifers are underappreciated hotspots for numerous untapped microbial immune resources. In this study, the detected number of microbial phyla (137 bacterial and 15 archaeal phyla) hosting defense systems is three folds higher than those reported in the NCBI RefSeq database (Fig. 1C, see "Methods"), in which each phylum displays distinct carriage patterns of defense system (see Supplementary Fig. S10 for patterns of each phylum). Although a few defense families are dominant with a continuous and dense distribution in either bacterial or archaeal genomes, the majority occurs with a patchy distribution (see Supplementary Fig. S11 for distribution of each defense system) as indicated by the first-line and accessory immune hypothesis.

Among the 27,578 bacterial MAGs, the Pseudomonadota phylum has the highest number of MAGs (9275), followed by CPR (3004 MAGs), the phylum representing ultrasmall bacterial symbionts in aquifers (Supplementary Figs. S2B, S10A, B and Supplementary Data 6). Compared with other microbial lineages (Fig. 2A, B), CPR genomes possess much fewer defense systems per genome (Wilcox test, Bonferroni-adjusted *P* < 0.0001), likely due to their streamlined genomes that restrict the ability to bear the fitness costs of accumulating defense systems[32,34,50], especially in the nutrient-constrained groundwater environment[51]. These ultrasmall prokaryotes might benefit from the rapid evolution of their immune arsenals by dynamically acquiring and losing systems from the "pan-immune pool", thereby

maximizing the defense capacity and benefiting the whole community[6,52]. Much higher defense system density (Wilcox test, Bonferroni-adjusted $P < 0.0001$) is observed in CPR than in Pseudomonadota and other phyla (Fig. 2B), suggesting that symbiotic CPR might undergo consistent viral infection and interplay with viruses frequently, as our previous study found[37].

Further, defense systems across major microbial lineages were characterized in 21 dominant bacterial phyla (with ≥100 MAGs) (Fig. 2C–E and Supplementary Fig. S12). The distribution of defense systems varies largely among distinct bacterial phyla (Supplementary Fig. S11B), which is attributed to differences in bacterial pan-genomes shaped by different lifestyles[53]. The diversity of defense families in four bacterial phyla (Campylobacterota, CPR, Nitrospirota, and Nitrospinota) exceeds the average across all 21 phyla (Fig. 2E). Campylobacterota, known as chemoautotrophs, can utilize sulfur, hydrogen, arsenic, and nitrate[54]. A rich arsenal of defense systems in Campylobacterota (Fig. 2E and Supplementary Fig. S12A) implies the immune adaptation of primary producers to evade viruses in groundwater. Similarly, Nitrospirota and Nitrospinota, both involved in nitrite oxidation and often found in marine and groundwater environments, also show a rich defensome linked to their chemolithoautotrophic lifestyles[55,56]. Myxococcota, notable for their complex multicellular community behaviors[57] (predate other microbes through "wolf-pack attacks"[58]) and large genome size (Fig. 2C), exhibit a distinct immune strategy associated with their predatory lifestyles[59], with the highest diversity of defense families per MAG (Fig. 2D and Supplementary Fig. S12E). The prevalence of AbiE systems in Myxococcota (Fig. 2F) further highlights their unique defensome, in which abortive infection promotes community survival by inducing rapid cell death to prevent viral release[10,43].

Unique defensome characteristics are observed in archaea. The DPANN superphylum, small cell symbionts similar to CPR bacteria[32], display significantly lower defense system number, density, and diversity than the Euryarchaeota superphylum (Supplementary Fig. S4E–G). The relationship between microbial lifestyles and their defensome seems more complex in DPANN, despite their similar physiological traits and close phylogenetic relationship with CPR[60]. In addition, Methanobacteriota, as an archaeal phylum with methanogenic metabolism, carries greater proportions of R-M and CRISPR-Cas but lacks AbiE (Supplementary Fig. S4H), suggesting a defense strategy focusing on degrading exogenous genetic materials rather than inducing cell death. It is worthy of noted that the observed 'low defense system density' in certain clades (e.g., CPR, DPANN) might also reflect the presence of exotic or unknown defense systems not currently detectable by existing methods.

This study provides a comprehensive assessment of the defensome of prokaryotes in groundwater. While the well-established bioinformatic tools (annotation methods with stringent filtering criteria)[18] are employed with carefully selected parameters to ensure the reliability of results, some degree of false positives or misclassification could still not be avoided[18,23]. The evolving nature of defense system discovery[1,8,13,16,17] and refinement of detection tools[42] may also lead to updates in specific annotations over time. Nevertheless, we expect the general trends observed in this study will remain robust, offering valuable insights into the defensome of environmental microbes.

## Defensome of small microbial symbionts under intense phage infection

The microbial defensome is shaped not only by the physiological and ecological traits of different lineages but also by the intensity of viral infection[40]. Based on the same metagenomic data, we investigated the effects of viral infection on defensome in groundwater by integrating virus-host ratios (VHR) from our previous study[37] (Supplementary Data 7). Both the density of defense systems and the diversity of

defense families significantly correlate with the VHR (Fig. 3A–C), suggesting a potential positive feedback mechanism that intense phage infection drives the expansion and diversification of microbial defensome. While the weak to moderate Spearman's rho values necessitate the need for additional omics or experimental evidence to substantiate this hypothesis, recent studies provide complementary support. Experimental evidence indicates that the number of defense systems in a bacterial strain could affect the level of susceptibility to already infecting phages[61]. And a significant positive correlation ($r = 0.70$, $P < 0.0001$) between defense system abundance and phage abundance was observed in the activated sludge system[62].

Our previous study has found that CPR and DPANN (ultrasmall symbionts in groundwater) are struggling with intense phage infection[37]. To investigate how these lineages adapt to such intense infection, we focused on their unique defensome profiles. Within CPR, the Paceibacteria and Microgenomatia classes emerge as the main targets of viruses[37]. Consistently, these classes exhibit a similar defensome composition, clustering closely and displaying a higher diversity of defense families compared to other classes (Fig. 3D, E). Paceibacteria shows the highest defense system density, and Microgenomatia also has a density above the average, underscoring their robust immune arsenals (Fig. 3E). A similar pattern is also observed in the DPANN superphylum. Nanoarchaeota and Aenigmatarchaeota are the most virally targeted phyla[37], clustering together and possessing a higher diversity of defense families (Fig. 3F, G). This shared vulnerability to viral infection, coupled with diverse defense arsenals, suggests a critical role for these defense systems in protecting these ultrasmall symbionts from intense phage pressure.

## Trade-offs between defense systems and adaptive traits

In integrative and conjugative/mobilizable elements (ICEs/IMEs)[63], a previous study has indicated the trade-offs between defense systems and antibiotic resistance genes (ARGs). Here, we find that such trade-offs can be extended to the whole prokaryotic genomes, evidenced by the inverse relations between the number of defense systems and the acquisition of adaptive traits, such as ARGs, heavy-metal resistance genes (MRGs), and virulence factor genes (VFGs) in microorganisms of groundwater environment (Fig. 4A). And a threshold effect is observed with a logistic-like curve that shows a dramatical decrease in the number of ARGs (Fig. 4B) or other traits (Supplementary Figs. S13 and S14) once the number of defense systems exceeds certain value. These defense systems are further categorized into first-line and accessory systems, in which the former exhibits a much stronger inverse correlation than the latter with ARGs. Among the five first-line systems (Fig. 4C, D), the AbiE has the strongest suppression of ARGs (Supplementary Data 8). These findings suggest that first-line defense systems are more tightly linked to reductions in adaptive traits (Fig. 4E, F) due to their pivotal role in the microbial immune landscape. Similar trends are observed for other adaptive traits like MRGs and VFGs (Supplementary Figs. S13 and S14, Supplementary Data 8), implying the microbial evolutionary balance between maintaining a robust immunity and acquiring external adaptive advantages. On one hand, these immune systems (e.g., CRISPR) protect bacteria and archaea against viruses and other mobile genetic elements (MGEs)[24,63], thus curbing the acquisition of adaptive traits carried by other MGEs, such as plasmid-borne ARGs[64–66]. On the other hand, bacteria with limited defense are likely to be infected by these genetic elements, which allows frequent HGT of resistance genes and virulence factors from genetic elements to bacterial genomes. Here, the underlying trade-off mechanisms could help to interpret the corresponding characteristics of microbial adaptive traits under varying virus-host interactions. For example, a recent study observed the natural water "self-purification"

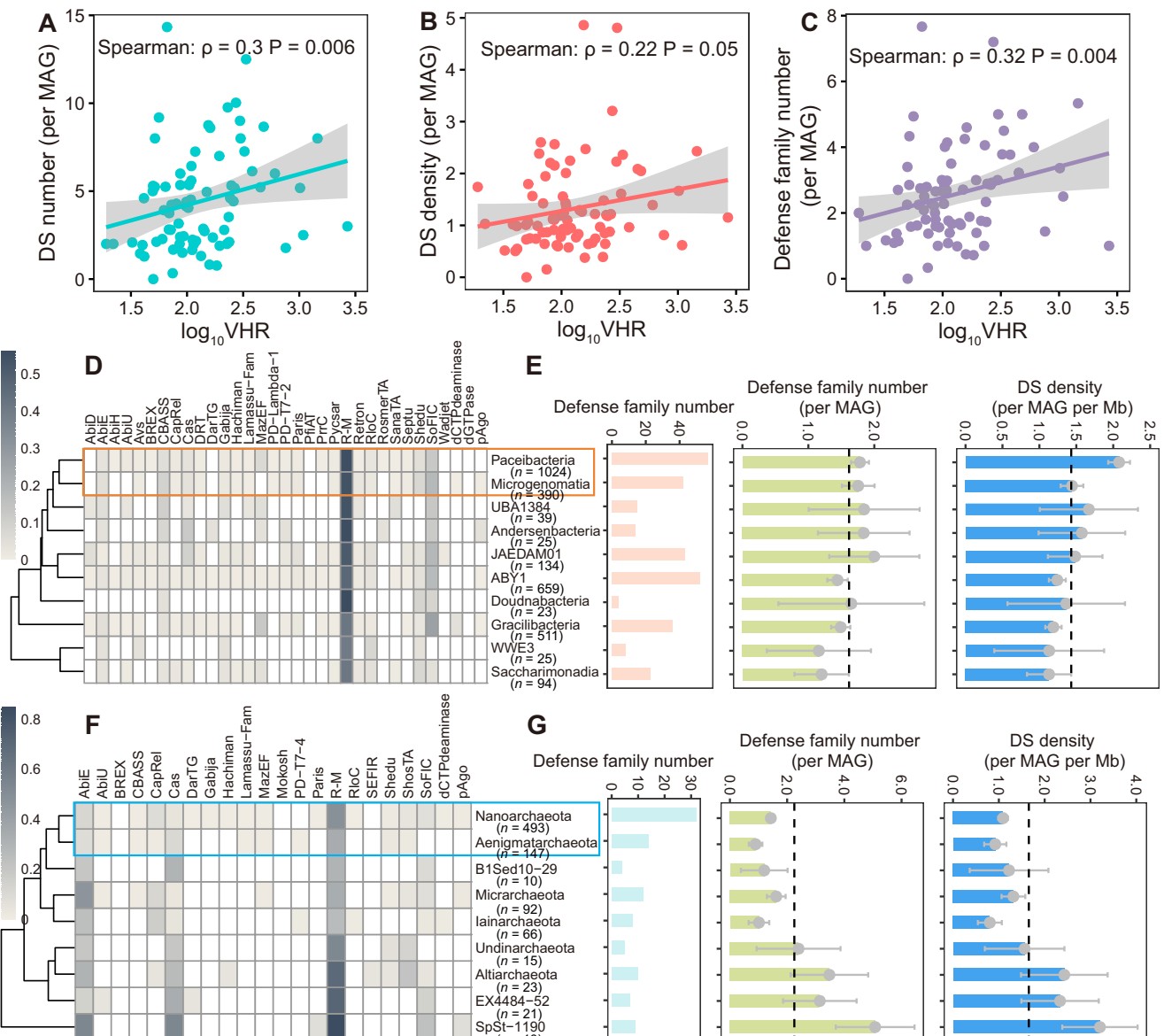

**Fig. 3 | Relationship between viral infection intensity and the prokaryotic defensome. A** Positive correlation between the defense system (DS) number and virus-host abundance ratios (VHR). **B** Relationship between defense system density (per MAG) and VHR. **C** Relationship between defense families (per MAG) number and VHR across all groundwater prokaryotic phyla. In (**A**–**C**) the central line was added by using "lm" method, with 0.95 confidence intervals marked by gray shading. Correlation was assessed using a two-sided Spearman's rank test with exact P-value provided. **D** Frequency of defense systems across CPR classes. Heatmap colors represent the presence frequency of each system (columns) in the genomes of each CPR class (rows). To improve visualization, only defense families present in at least 1% of all CPR genomes and classes with at least 20 MAGs are

shown. The number of MAGs analyzed are shown as *n* values. **E** Total defense families, defense families per MAG, and defense system density (per MAG and per Mb) across CPR classes. Error bars represent the 95% confidence interval (CI) of the mean. The number of MAGs analyzed are shown as *n* values. **F** Frequency of defense systems across DPANN phyla. Heatmap colors represent the presence frequency of each system (columns) in the genomes of each DPANN phylum (rows). Only defense families present in at least 0.5% of all DPANN genomes and phyla with at least 10 MAGs are shown. **G** Total defense families, defense families per MAG, and defense system density (per MAG and per Mb) across DPANN phyla. Error bars represent the 95% confidence interval (CI) of the mean. The number of MAGs analyzed are shown as *n* values. Source data are provided as a Source Data file.

phenomenon[67], where pathogens carrying abundant ARGs are suppressed by phages. This could be partially explained by trade-offs between defense systems and ARGs.

As ARGs continually pose a significant challenge to human and environmental health, phage therapy has emerged as a promising alternative to combat antibiotic-resistant bacterial infections[68–70]. In this regard, the trade-offs between defense systems and adaptive traits strengthen the potential of using phages to inhibit super-resistant bacteria, because these bacteria often possess limited anti-MGE capabilities, rendering them more susceptible to phage-mediated elimination[21].

## Genetic mobility and genomic co-localization of defense systems

We also investigated the roles of major families of MGEs (plasmids, phages, prophages, integrons, and ICEs/IMEs) in the mobility of defense systems[71]. Previous studies in other ecosystems have shown that MGEs can carry defense systems, promoting their spread for selfish propagation[47]. In groundwater ecosystems, we identified 163,897 plasmids, 40,302 phages/prophages, 2697 integrons, and 606 ICEs/IMEs across 27,578 bacterial MAGs, with the highest representation in Pseudomonadota (Fig. 5A, see Supplementary Fig. S15 for detailed distribution of MGEs). Notably, CPR exhibits the highest

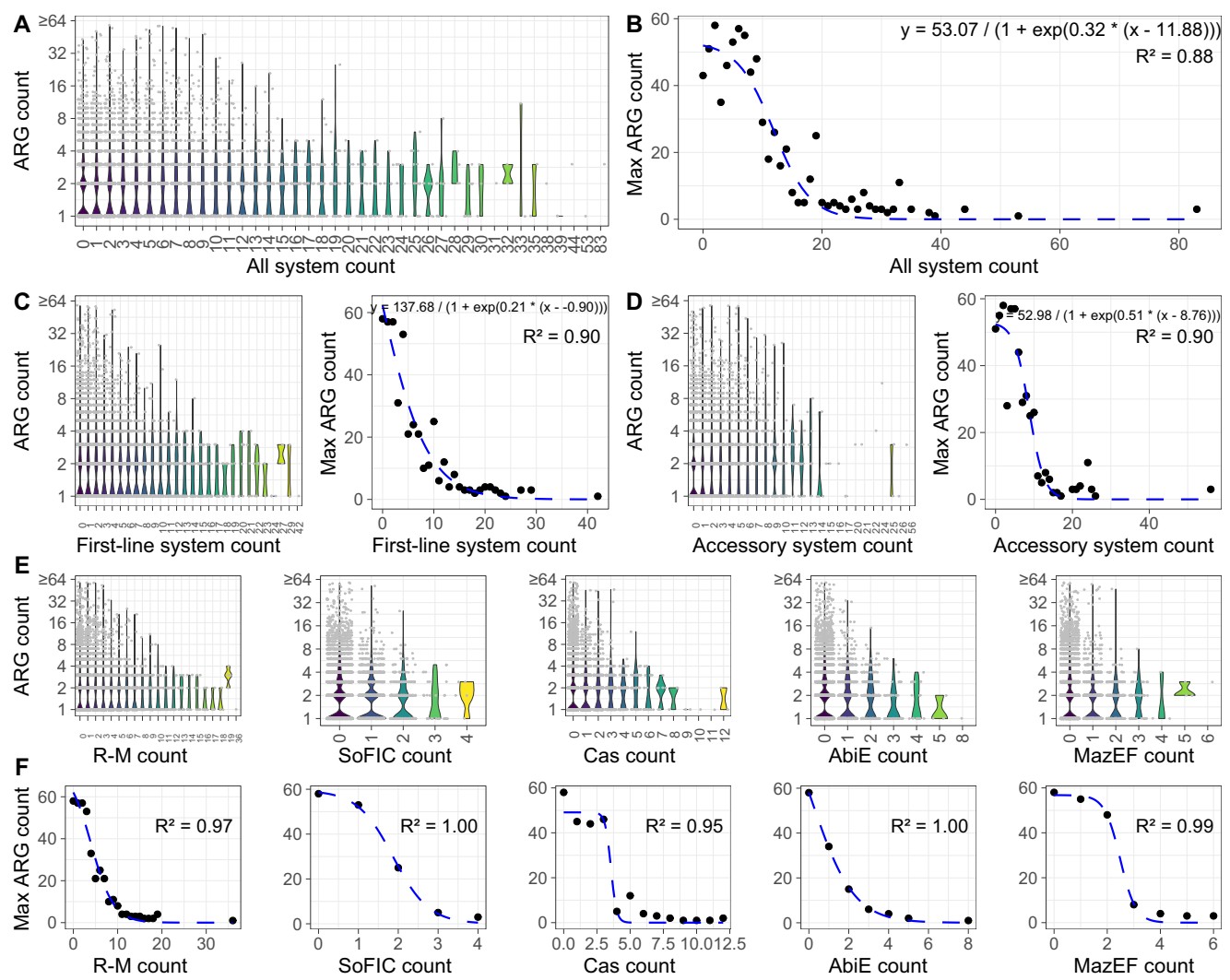

**Fig. 4 | Inverse relationship between defense systems and other adaptive traits.** Only antibiotic resistance genes (ARGs) are shown herein, other traits are shown in Supplementary Figs. S12 and S13. **A** Change in ARG numbers with increasing defense system numbers across all bacterial genomes. **B** Decrease in the upper limit of ARG numbers as defense system numbers increase. **C** Relationship between ARG numbers and first-line defense system numbers. **D** Relationship between ARG numbers and accessory defense system numbers. **E, F** Relationship between ARG numbers and the number of each first-line defense system (RM, SoFIC, CRISPR-Cas, AbiE, and MazEF). See Methods for detailed information of fitting models. Source data are provided as a Source Data file.

proportion of phages/prophages (73% of their MGEs, Fig. 5B), aligning with the finding that they are under intense viral infections[37]. Defense systems are more densely localized on MGEs compared to chromosomal regions (Fig. 5C), with all types of MGEs carrying a higher-than-expected number of defense systems from a wide range of defense families (Fig. 5D). Unlike first-line defense systems such as R-M, SoFIC, CRISPR-Cas, and MazEF, many accessory immune systems are preferentially positioned on MGEs, suggesting that microorganisms dynamically adapt to their environments by horizontally acquiring or discarding these accessory systems[6]. In addition, distinct localization preferences among defense families are evident, with systems like BstA more commonly found on phages/prophages, while dGTPase tends to localize on plasmids (Fig. 5D). These findings highlight the complex interplay between defense systems and MGEs, with specific families favoring certain MGEs for their dissemination and maintenance[52].

Defense systems usually cluster within specific genomic regions known as defense islands[14] that serve as hotspots for discovering novel defense systems[13]. Using previously defined criteria for detecting these regions[22] (see "Methods"), we identified 669 defense islands in groundwater prokaryotes: 662 from 27,578 bacterial MAGs and 7 from 1453 archaeal MAGs. In comparison, defense islands are more

abundant in soil, marine, and human gut[22] environments than in groundwater. Several factors may explain this disparity: (1) Streamlined genomes. Groundwater microorganisms, particularly CPR and DPANN, have streamlined genomes with limited metabolic flexibility and transient defense systems adapted to the resource-constrained environment, which may limit the formation of defense islands[72,73]. (2) Reduced HGT. Similar to marine environments[22], the low cell density and planktonic lifestyle in groundwater could restrict HGT events, thereby reducing the formation of defense islands.

In bacterial genomes, 61.4% of genes (4538 of 7391) in 662 defense islands are annotated as defense genes, encompassing 2096 defense systems of 98 defense families. First-line defense families (i.e., R-M, AbiE, and SoFIC) comprise only 39% of these systems. The remaining systems belong to a diverse set of accessory immune families, accounting for 61% of defense systems in defense islands (Fig. 5E). Interestingly, the five first-line defense families (R-M, SoFIC, CRISPR-Cas, AbiE, and MazEF) don't show preferential localization in defense islands, whereas many accessory defense families are significantly enriched in defense islands (Fig. 5I), consistent with findings on the co-localization pattern of defense systems and MGEs (Fig. 5D). Hence, defense islands may function as the accessory immune reservoirs, where dynamic and

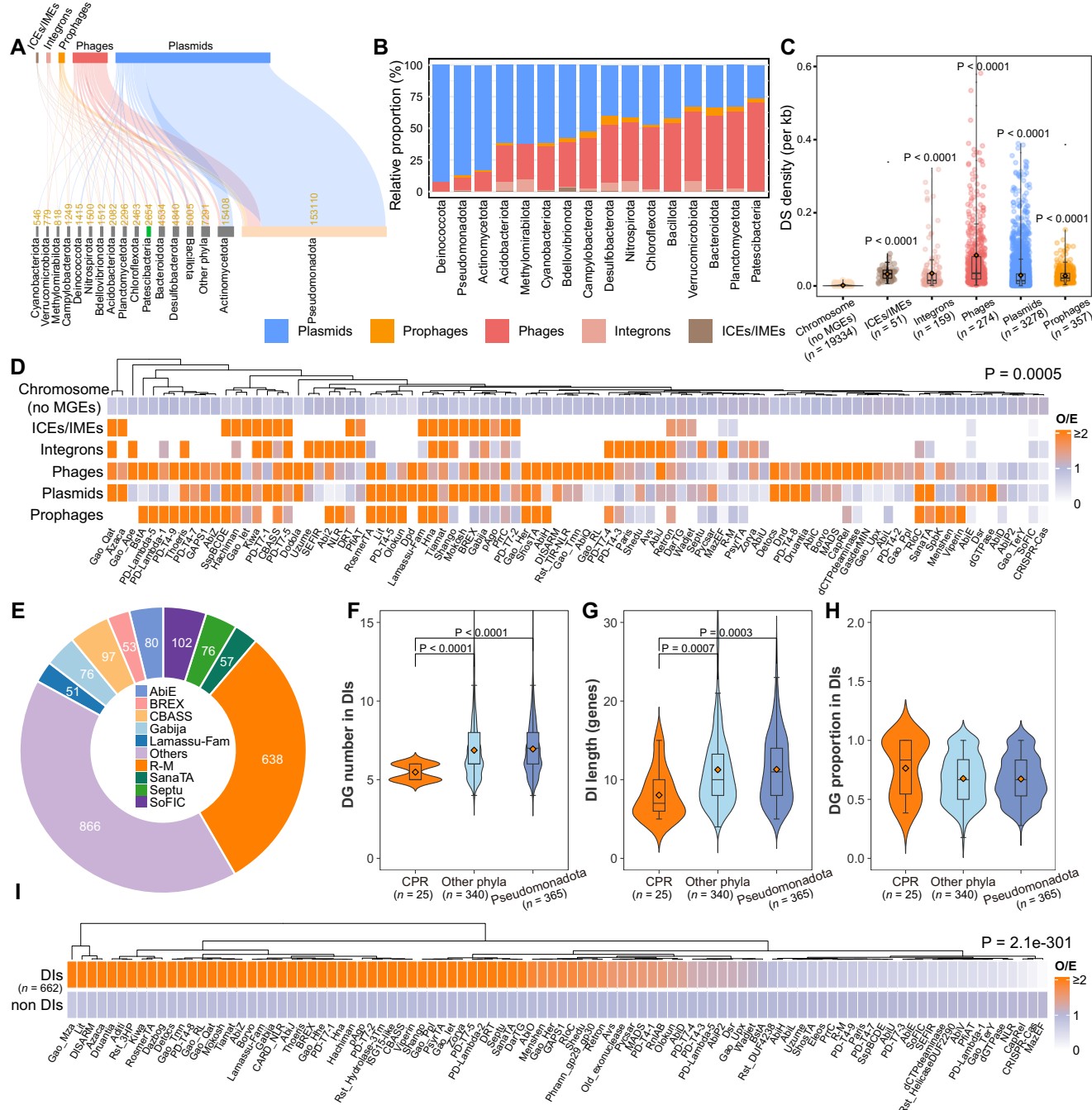

**Fig. 5 | Mobile genetic elements (MGEs) as carriers of accessory defense systems and their defense island reservoir. A** Distribution of bacterial phyla of MGE hosts. **B** Relative percentages of MGEs across phyla, showing only phyla containing ≥ 500 MGEs. **C** Comparison of defense system (DS) density (per kb) between chromosomes and other MGEs (two-sided Wilcox test, Bonferroni-adjusted P-values). The number of MGEs analyzed are shown as *n* values. **D** Heatmap of observed/expected (O/E) ratios for colocalization of DSs and MGEs. Expected values were calculated by multiplying the total number of systems in the given defense family by the fraction assigned to the chromosome, or the other MGEs. Significance was assessed by a two-sided Chi-square test. **E** Number of defense systems across different defense families, with families having < 50 systems grouped as "Others". **F–H** Comparison of

defense genes (DGs) per defense island (DI), defense island length (genes), and proportion of defense genes within defense islands across CPR, Pseudomonadota, and other phyla (two-sided Wilcox test, Bonferroni-adjusted P-values). The number of DIs analyzed are shown as *n* values. **I** Heatmap of O/E ratios for colocalization of defense systems and DIs. Expected values were calculated by multiplying the total number of systems in a defense family by the fraction assigned to DIs or non-DIs. Significance was assessed by a two-sided Chi-square test. Boxplots in (**C**, **F–H**) represent the 25th to 75th percentiles, the inner black line marks the median, whiskers extend to 1.5x the interquartile range, and the orange square dot represents the mean. Source data are provided as a Source Data file.

horizontally transferable systems accumulate. In the CPR bacteria, 25 defense islands are identified with fewer defense genes compared to other microorganisms (Fig. 5F, G), but the proportion of defense genes is relatively higher within their islands (Fig. 5H), indicating the importance of defense in these streamlined genomes.

Apart from defense genes, the remaining 2,853 genes (see Supplementary Fig. S16 for functional annotations) in defense islands may play auxiliary roles in the defense process or represent novel systems. These genes were classified into 36 COG categories, including S (function unknown, 480 genes), L (replication, recombination, and

repair, 383 genes), K (transcription, 238 genes), V (defense mechanisms, 137 genes), and others (316 genes). Genes annotated as L, K, and V play important roles in cellular information storage and processing, cellular processes and signaling[74]. Their prevalence (Supplementary Fig. S16A) suggests a functional link to microbial immunity, potentially representing unidentified defense genes. KEGG and PFAM annotations (Supplementary Fig. S16B, C) further highlight defense-related domains, such as type I restriction-modification DNA specificity domain (K01154), ATP-dependent DNA helicase (K03655), and WYL transcriptional regulators[75,76], underscoring their connection to microbial defense[22]. In archaeal genomes, we identified 7 defense islands containing 22 defense systems across 9 defense families, with R-M showing the highest prevalence (11 of 22) as observed in bacterial genomes.

These findings reveal the connection between defense systems, MGEs, and defense islands, with an emphasis on less abundant but highly diverse accessory immune systems. The co-localization patterns of defense systems in MGEs are highly consistent with those in defense islands. Accessory immune systems are more frequently co-localized within both MGEs and defense islands, while first-line systems exhibit rare co-localization (Fig. 5D, I). This suggests that defense islands serve as hotspots for the MGE-mediated dissemination of accessory immune systems in groundwater[52,77]. Our findings also reveal additional offensive tools (type IV secretion system[78] and other secretion systems[79]) in defense islands (see Supplementary Fig. S17 for their genomic contexts), which may support the concept that "defense island" should be broadened to include both defensive and offensive tools as suggested in a recent study[80].

### Phage anti-defense and interaction between CRISPR-Cas and anti-CRISPR

In response to the diverse prokaryotic defensome, phages have evolved a variety of anti-defense mechanisms to ensure successful infection[81]. These anti-defense genes play a crucial role in the ongoing arms race between prokaryotes and their phages, driving the diversification of both prokaryotic defense systems and viral counter-strategies[82,83]. To explore phage anti-defense mechanisms in groundwater, we identified 712 anti-defense genes across nine distinct anti-defense types from 625 phages, accounting for only 1.6% of all 40,302 phages extracted from 27,578 bacterial MAGs (see Supplementary Fig. S18 for the distribution of anti-defense genes). Here, we find the most prevalent anti-defense types are anti-Thoeris (27.7%), anti-CRISPR (27.4%), and anti-RM (26.2%) (Supplementary Fig. S18A). The high prevalence of anti-CRISPR and anti-RM correlates with the high abundance of CRISPR-Cas and R-M systems in bacteria. We also find the co-localization of anti-defense genes within phages (Fig. 6A), which may enhance viral infection capacity. More importantly, this co-localization may lead to the discovery of new viral anti-defense mechanisms in the vicinity of co-located anti-defense genes, akin to the discoveries of novel defense systems using a similar way[84]. 65 phage contigs are found to harbor both anti-defense genes and defense systems (Supplementary Fig. S18C). This suggests that phages may utilize anti-defense mechanisms to parasitize their hosts while employing defense systems to combat other competing MGEs, thereby optimizing their survival and dissemination[85,86].

To gain insights into historical virus-host interactions, we mined CRISPR spacers from bacterial MAGs that contain Cas proteins[87]. By matching these spacers to protospacers (viral genomic regions targeted by the CRISPR-Cas system) in phages, we inferred bacterial adaptive immunity against 601 phages, 43 of these phages carry a total of 54 anti-defense genes (Supplementary Fig. S18B): anti-CRISPR (also known as Acr, 27), anti-RM (8), anti-Thoeris (8), anti-CBASS (6), and anti-Dnd (5). The dominance of anti-CRISPR is expected, as these phages targeted by spacers need corresponding anti-defense mechanisms to evade the CRISPR-Cas system. The presence of other anti-defense types suggests potential cross-defense interactions, where bacteria may use CRISPR-Cas to eliminate phages that carry other types of anti-defense genes but no anti-CRISPR, thereby circumventing their countermeasures (Supplementary Fig. S18B) and promoting the diversification of viral anti-defense strategies.

We further investigated the diversity of viral Acr proteins by extracting 41 Acr genes from phages targeted by spacers and 339 experimentally validated Acr genes from the PaCRISPR database to construct the phylogenetic tree (Fig. 6B). These predicted Acr proteins display broad distribution across different subtypes, with diverse structures confirmed by computational modeling (see Supplementary Fig. S19 for predicted structures). This aligns with a previous study on the diversity of Acr proteins[88]. Notably, spacer-matched virus-host links reveal the inhibition of a phage-encoded Acr protein targeting a newly found complete CRISPR-Cas9 system in *Rugosibacter sp002422995* (Fig. 6C, D, see Supplementary Fig. S20 for multiple sequence alignment of Cas9), showing how Acr proteins are employed by phages to counteract the CRISPR-Cas in their host. Given that CRISPR-Cas9 is widely used in gene editing, this interaction could offer a new regulatory perspective for gene editing applications, opening up potential biotechnological advancements based on CRISPR-Acr interactions[89].

In summary, our study significantly expands the knowledge pool of defense systems by providing a catalog of prokaryotic defensome in groundwater. The entire microbial defensome pattern is well depicted based on first-line and accessory defense systems. CRISPR-Cas plays an important role in defense in many environments, while our newly discovered complete CRISPR-Cas9 from *Rugosibacter sp002422995* and its interaction with Acr may offer new tools for revolutionizing the gene editing process. Due to the ongoing arms race between bacteria and phages, groundwater viruses also evolve a diverse range of anti-defense genes that target various defense strategies. In the future, more efforts are needed to improve our understanding of complex virus-host interactions by profiling the immune landscape of numerous uncultured microorganisms in the subsurface biosphere and other ecosystems.

## Methods

### Study area and sample collection

Metagenomic sequencing was conducted on 607 groundwater samples obtained from 525 newly established and 82 upgraded monitoring wells across China during 2016-2017 (Supplementary Figs. S1 and S2A) as mentioned in our previous study[37]. Briefly, the monitoring wells were distributed in seven distinct geological regions. The well depths varied between 0 m and 600 m, with diverse hydrological and geological contexts as well as groundwater burial conditions. Groundwater was pumped after flushing and filtered using 0.22 μm polycarbonate membranes (Millipore) to collect microorganisms. All membranes were then stored at −80°C for subsequent high-throughput sequencing. Given the exceptionally large volume (2000 L) of water filtered and our focus on broad microbial communities, the potential loss of some ultra-small cells (CPR/DPANN) due to the use of a 0.22 μm filter is considered an acceptable trade-off.

### DNA extraction and sequencing

Total DNA was extracted using the FastDNA SPIN Kit for Soil (MP Biomedicals, USA) following the manufacturer's instructions. The concentration and quality of the DNA were then measured using a NanoDrop spectrophotometer (NanoDrop Technologies Inc., Wilmington, DE, USA). Genomic DNA was sequenced using the Illumina HiSeq 4000 platform (Shanghai Majorbio company) with 2 × 150 bp paired-end reads. A total of 607 groundwater metagenomic samples were sequenced, with data sizes ranging from 30.3 to 52.2 gigabases and an average of 34.3 gigabases per sample.

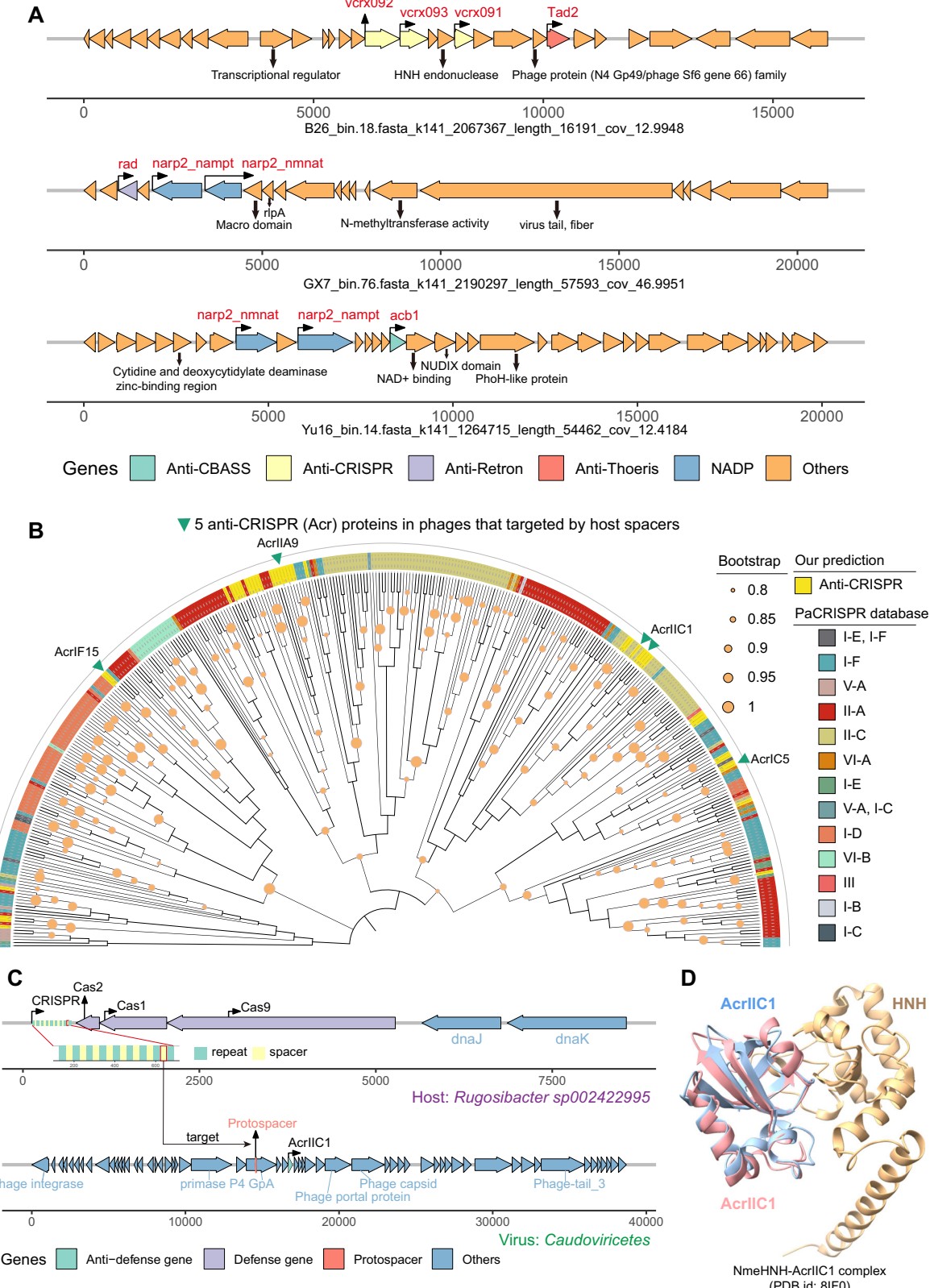

**Fig. 6 | Characteristics of anti-defense genes in phages. A** Co-localization of anti-defense genes within phages. **B** Maximum-likelihood phylogenetic tree of 41 predicted anti-CRISPR (Acr) proteins from bacterial genomes and 339 Acr proteins from the PaCRISPR database. Green triangles on the outer ring denote Acr proteins in phages that targeted by bacterial CRISPR spacers. **C** Interaction between a newly discovered CRISPR-Cas9 system and an Acr protein. The CRISPR spacer locus that targets the phage protospacer is highlighted in red. **D** Putative mechanism of AcrIIC1 interaction with the Cas9 HNH domain. The primary structure is the NmeHNH-AcrIIC1 complex (PDB id: 8IF0), with the pink region representing the AcrIIC1 identified in this study. AcrIIC1 was fitted to the complex using ChimeraX's Matchmaker function. Source data are provided as a Source Data file.

## Metagenomic sequencing data processing

The metagenomic data were processed using metaWRAP v1.2.3[90]. Quality control was performed on each sample using the metaWRAP Read_qc module with parameters: --skip-bmtagger. Clean reads from each sample were assembled using the Assembly module (--megahit -l 500). The assembled contigs were binned using the metaWRAP Binning module (--maxbin2 --concoct --metabat2). Generated metagenome-assembled genomes (MAGs) were then refined using the metaWRAP Bin_refinement module (-c 70 -x 10). The completeness and contamination of each genome were assessed using CheckM v1.1.2[91]. All MAGs were then dereplicated using dRep v2.5.4 (-pa 0.9 -sa 0.99 -cm larger -comp 75 -con 5 -nc 0.30)[92]. An average nucleotide identity (ANI) of 99% was selected to generate strain-level MAGs for the characterization of defense systems, as these systems usually have a pan-genomic feature and vary among different strains[19,20]. For example, new defense systems have been identified in the *E. coli* pan-genomes[93]. The optimal growth temperature of genomes was predicted using scripts from (https://github.com/DavidBSauer/OGT_prediction)[94]. Taxonomic annotation of dereplicated MAGs was performed using the classify_wf module (--mash_db) of GTDB-Tk v2.4.0 (dataset R220 version)[95], and MAGs were classified using the naming system of the GTDB taxonomy. To ensure fair comparisons, we also used the "gtdb_to_ncbi_majority_vote.py" script from the GTDB-Tk repository[95] to obtain NCBI classification annotations of these MAGs based on their positions in the GTDB tree. These annotations were only used for comparison with the NCBI RefSeq complete prokaryotic genome database. Given that defense systems can be multigenic, a stricter completeness threshold ($\geq 70\%$) was employed in this study compared to the conventional 50% threshold used in other metagenomic analyses[96,97]. Despite applying a higher completeness threshold, partially assembled genomes may still result in the incomplete recovery of multi-gene defense systems and certain defense islands. So, high-quality MAGs (completeness $> 90\%$, contamination $< 5\%$, completeness-5 * contamination $> 70$, N50 $> 100$ kb)[96] were used for analysis when compared with other studies and NCBI RefSeq to control for the potential influence of MAG quality on results. Plasmids and phages/prophages in MAGs were identified and classified using geNomad v1.8.0[71] (parameters: end-to-end). Integrons were identified using IntegronFinder v2.0.5[98] (--local-max --func-annot --gbk --pdf) and only complete integrons were retained in downstream analyses. Integrative Conjugative Elements (ICEs) and Integrative Mobilizable Elements (IMEs) were detected with ICEfinder v2.0[99] (-t Metagenome).

## Functional annotation

MAGs were queried for defense genes/systems using DefenseFinder v1.3.0[18] (--preserve-raw), a tool that detects known defense systems in prokaryotic genomes using a comprehensive collection of hidden Markov model (HMM) protein families and genetic organization rules targeting all major defense systems[18,22]. The defense system information of the NCBI RefSeq complete prokaryotic genome database used in this study for comparison was downloaded from the DefenseFinder webservice[16] (Supplementary Data 3). All defense gene sequences in RefSeq prokaryotic genomes were downloaded from NCBI based on accession number, resulting in a total of 198,355 sequences (Supplementary Data 3). These sequences were first used to construct a sequence library with makeblastdb. Next, all predicted defense genes in the groundwater prokaryotic MAGs were searched (-*e*-value 1e-5) against this database using blastp. The blastp results were then filtered using < 80% coverage and < 90% identity to identify sequence novelty of groundwater defense genes, considering these parameters are commonly employed in protein clustering analyses. The open reading frames (ORFs) of dereplicated MAGs were predicted using Prodigal v2.6.3[100]. Then, all ORFs were searched against SARG v3.2.1[101], BacMet v2[102], and the virulence factor database (VFDB)[103] using diamond v2.0.6 blastp (-e 0.00001 --id 60 --subject-cover 90) to annotate ARGs, MRGs,

and VFGs. Defense genes/systems on all MGEs were annotated using the same methodology for prokaryotic MAGs to assess the mobility of defense systems. DefensePredictor[42] was applied to the groundwater high-quality MAG set (1626 MAGs) to identify possible defensive genes using a stringent probability cutoff of > 0.999.

## Experimental validation of defense systems

R-M and SoFIC systems were identified as the most abundant defense systems in groundwater prokaryotes, representing first-line immunity (Fig. 1A), with type IIG R-M being the most prevalent R-M subtype. Therefore, type IIG R-M and SoFIC were selected for experimental validation of their anti-phage activity using plaque assays. In addition, the CBASS system, representing accessory immunity, was also selected for validation. For type IIG R-M and SoFIC, two gene variants were chosen for each system: one closely related to the corresponding HMM profile (i.e., the top homology candidates prioritized by hit score) and one more distantly related (randomly selected from low-homology candidates). Distant homologs were selected using the following median-based thresholds: Type IIG R-M—hit score < 668, hit_profile_cov < 0.94, hit_seq_cov < 0.94, hit_seq_len < 1044; SoFIC—hit score < 375, hit_profile_cov < 0.96, hit_seq_cov < 0.95, hit_seq_len < 373. In total, five defense systems were evaluated (see Supplementary Data 10 for detailed information). The coding sequences of the selected genes were commercially synthesized in GenScript and cloned into either the pQE-60 (type IIG R-M and SoFIC) or pETDuet-1 (CBASS) expression vector. Recombinant plasmids were transformed into *Escherichia coli* B (BL21), with empty vector controls included for comparison. Plaque assays were conducted following established protocols[104]. A single colony from a fresh LB agar plate was inoculated into LB broth containing ampicillin (50 μg/mL) and grown at 37 °C until reaching an OD600 of ~0.6. For the pETDuet-1 vector, protein expression was induced with 0.2 mM isopropyl β-D-1-thiogalactopyranoside (IPTG). After an additional hour of incubation at 16 °C, 1.5 mL of the bacterial culture was mixed with 12.5 mL of 0.6% LB top agar containing ampicillin (50 μg/mL) and IPTG (0.2 mM, for pETDuet-1 vector only), and the mixture was poured onto LB plates containing ampicillin (50 μg/mL). Each plate was then spotted with 5 μL aliquots of serially diluted phage suspensions ($10^6$ to $10^2$ for T1 and T7 phages). For statistical comparison, the plaque-forming unit (PFU) was counted at 7 h post-infection. Plates were incubated at 25 °C overnight before imaging. All experiments were performed in three batches for replicability.

## Defense islands identification

For the sake of comparability, the definition of defense islands from a previous study[22] was used here, i.e., arrays of defense genes separated from one another by 10 genes or less and containing at least five genes belonging to a minimum of three different defense families. Using this definition, we identified defense islands by extracting genomic regions from the MAGs that met these criteria. In addition, functional annotation for non-defense genes in defense islands was performed using eggNOG-mapper v2[105] (--override -m diamond --pident 40) against precomputed eggNOG v5.0 databases[106]. We also searched the conceived domains (CDs) in these sequences using NCBI CD-search (concise mode, *e*-value 1e-5) against the CDD database v3.21[107].

## Phage anti-defense identification

To further analyze virus-host interactions, anti-defense genes in phages were identified using DefenseFinder (--antidefensefinder-only --preserve-raw)[108]. Note that DefenseFinder HMMs are primarily developed for major anti-defense systems (e.g., anti-RM, anti-CRISPR)[108], and non-first-line systems either lack reliable HMMs or exhibit poor accuracy, it is far from sufficient for understanding the anti-defense landscape based on the predicted results because of the limitation of current knowledge on phage anti-defense systems. To

validate our predicted anti-CRISPR genes, 339 experimentally validated Acr protein sequences from the PaCRISPR database[109] were downloaded to perform the phylogenetic analysis.

### Spacer extraction and protospacer-to-spacer matching

MAG contigs containing Cas genes were extracted, and CRISPR arrays in these contigs were identified using minced v0.4.2[110] (-spacers). Identified spacers in these contigs were then searched against the phage sequences using blastn to find protospacers in phages, with a mismatch ≤1 and coverage ≥95% as filtering parameters. Those phages targeted by spacers were extracted to predict their anti-defense genes using DefenseFinder (--antidefensefinder-only --preserve-raw)[108]. The protein structures of the anti-CRISPR genes and newly found CRISPR-Cas9 were predicted using AlphaFold3[111] and were visualized using ChimeraX[112]. To visualize the domain characteristics of the newly found Cas9 protein, this Cas9 protein was aligned with other previously identified Cas9 proteins from the *Rugosibacter* genus and COG3513 sequences from the CDD database using Clustal Omega[113]. The generated multiple sequence alignment was visualized using Jalview v2.11.4.1[114].

### Phylogenetic analysis

The phylogenetic tree of prokaryotic MAGs was constructed using the infer module of GTDB-Tk based on the multiple sequence alignments of 53 archaeal and 120 bacterial markers. Phylogenetic depth was calculated as the diagonal mean of the phylogenetic variance-covariance matrix for phylogenetic tree using the *vcv.phylo* function in R package *ape*. The multiple sequence alignments of anti-defense genes predicted from groundwater phages and 339 experimentally validated Acr proteins were generated using Muscle v5[115] and trimmed with trimAl[116] (-gt 0.5). Maximum likelihood phylogenetic trees of these sequences were generated using FastTree v2.1.11[117], and visualized using Interactive Tree of Life (iToL) v6[118].

### Abundance profiling

Reads per kilobase per million mapped reads (RPKMs) were used to estimate the relative abundance of genomes. Clean reads were mapped to the genome set using Bowtie2[119]. Mapped reads were processed with SAMtools[120] for sorting. The sorted BAM files were then analyzed with CoverM v0.3.1 (https://github.com/wwood/CoverM) to exclude low-quality mappings and create RPKM profiles for all samples.

### Statistical analyses and visualization

Statistical analyses were carried out in R v4.3.1 (https://www.R-project.org/). Normality of data was evaluated using the Shapiro–Wilk tests before statistical analyses. Two-sided Spearman correlation analysis was used to test the relationship between defense system number/density and genome size. Wilcox test (Bonferroni-adjusted $P$-value) was used to test the difference in defense system number/density between CPR, Pseudomonadota, and other phyla. Stepwise linear regression analyses were performed to examine the relationship between the number of defense genes/systems, genome size, phylogenetic depth, and N50 using the *lm* and *step* functions in the *stats* package. Genomic contexts of defense islands, defense system co-localization, and anti-defense gene co-localization were visualized using the *gggenes* package. The observed/expected (O/E) radio was calculated using *chisq.test* function. The upper and lower sides of all boxes represent the interquartile range between the 25th and 75th percentiles. Fitting models of maximum ARG/MRG/VFG number and defense system number were performed using the *nls* function and the following equations, analogs of the sigmoid function.

$$y = \frac{a}{1 + e^{b*(x-c)}} \tag{1}$$

Where a, b, and c are unknown parameters to be fitted using our data, the fitted curve is symmetric about the point (c, a/2). The parameter b represents the steepness of the curve at the symmetric point. The fitting models were evaluated using $R^2$, calculated according to the following formula.

$$R^2 = 1 - \frac{\sum \left(y_{obs} - y_{fit}\right)^2}{\sum \left(y_{obs} - \bar{y}_{obs}\right)^2} \tag{2}$$

### Reporting summary

Further information on research design is available in the Nature Portfolio Reporting Summary linked to this article.

## Data availability

The metagenomic groundwater prokaryotic defensome catalog (GPDC), viral anti-defense genes, phylogenetic trees of anti-CRISPR (Acr) genes based on amino acid sequences, Acr sequences, newly found CRISPR-Cas9 sequences, and protein structures are available in the Zenodo repository (https://doi.org/10.5281/zenodo.13943877). All sequencing reads used in this study have been deposited in the NCBI database under accession code PRJNA858913. Source data are provided in this paper.

## Code availability

Shell scripts and R codes used in this study were publicly available on Zenodo (https://doi.org/10.5281/zenodo.15656571)[121].

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

## Acknowledgements

This work was supported by the National Natural Science Foundation of China (U2240205 and 51721006). Sincere thanks are to Dr. Yang Bai from the School of Life Sciences, Peking University, for his help in experiments. Bioinformatic support from the High-performance Computing Platform of Peking University and the open access policy of the Geocloud Database developed by the China Geological Survey are also acknowledged.

## Author contributions

J.R.N. designed the research. P.W.L. conducted the bioinformatic and statistical analysis with the help of Z.Z.W., T.L., and C.F.D. P.W.L. and Q.L. conducted the plaque assay experiments. P.W.L. wrote the manuscript, and J.R.N. revised the manuscript. All the authors read and approved the final manuscript.

## Competing interests

The authors declare no competing interests.
