## [Transparent Peer Review file · Nature Communications]

The defensome of prokaryotes in aquifers

Corresponding Author: Professor Jinren Ni

Version 0:

Reviewer comments:

Reviewer #1

(Remarks to the Author)

Li and co-workers propose a large-scale analysis of the groundwater prokaryotic anti-phage defensome. This is a timely work that follows the trend of a growing number of studies focusing on the environmental defensome. I have a few comments/suggestions that I hope the authors will find useful.

- 1) The recent use of machine learning models applied to the prediction of novel defense systems (<https://www.biorxiv.org/content/10.1101/2025.01.08.631726v1>) shows significant improvement over DefenseFinder concerning the number of novel families detected. Maybe it would be appropriate to see if more recent tools provide a more fine-grained view of the defensome and shift any of the conclusions?
- 2) The current know-how on phage anti-defense systems is still very limited, and the majority of DefenseFinder HMMs have been obtained for the major anti-defense systems (anti-RM, anti-CRISPR, etc), while non-first-line systems' HMMs are still poorly accurate or not built. Your 1.6% value (625 in 40,302 phages) demonstrates how far you are to have a solid view on the panoply of anti-defense systems. I would either remove the section on anti-defense, or caution the reader that this section risks to be highly inaccurate and poorly representative of the actual anti-defense landscape.
- 3) Did you observe any effect of groundwater sampling depth (until 600 m according to your previous study) on the defensome composition?
- 4) We have now accumulating evidence of the role of these defense systems against multiple classes of MGEs (and not only phages). Maybe change the term "antiphage" to "anti-MGE" across the text?
- 5) Fig. 5C Are these prophages? If not, the authors should disentangle between prophages and non-prophages. Does the 'chromosome' box plot contain MGEs? If so, they should control for this, by removing chromosomal MGEs.
- 6) What about ICEs/IMEs? Integrons? These are really important for the analyses.
- 7) Fig.1 caption: "from a previous study". Which one?
- 8) Page 7: line 157. This sentence comes in contrast to dozens of papers in the literature that support HGT as a main source of shuttling and accumulation of defense systems (notably in Dis). Maybe clarify?
- 9) The authors should caution for the fact that a "low defense system density" (or vice-versa) in a particular clade (e.g. CPR, DPANN) might arise simply by the fact that such clades harbor more exotic or unknown DSs not covered by the existing detection methods (or inversely, are very well covered by the existing methods).
- 10) Page 10: section adaptive traits. Wouldn't this inverse trend be already expected? If one has more defense systems, in theory, you should curb HGT, and therefore the acquisition of adaptive traits as the ones mentioned. Such "intriguing" finding as mentioned by the authors, seems actually expected, and was already described in the literature: <https://academic.oup.com/nar/article/51/9/4385/7132345>
- 11) Avoid acronyms in the abstract (e.g. CPR). Also use "R-M" instead of "RM" to agree with the official notation.

12) It would be fair if the authors could acknowledge the work that first introduced the term defensome applied to antiphage defenses (<https://www.nature.com/articles/s41467-024-46489-0>).

13) Page 5: Please avoid using DefenseFinder family nomenclature with underscores.

14) Please avoid the term “cornerstone” defense systems. For the sake of uniformity, I would suggest the term already in use in the literature “first-line” defense systems.

15) Please provide the wrapper scripts for all the analyses. These should be made public for questions of clarity.

Reviewer #2

(Remarks to the Author)

Li and coauthors carried out a robust survey of bacterial and archaeal defense, and viral anti-defense systems in genomes reconstructed from the groundwater biosphere. The manuscript describes a systematic effort to generate a compendium of genetic elements that have potential function in defense for their hosts, or separately, for viral resistance to host defenses. However, I find two major issues in review of the manuscript. First, the study lacks additional omics or experimental data (metatranscriptomes, proteomes, or genetic experiments) that could demonstrate that at least some of these defense systems are expressed, or even function as predicted. These defense systems are best characterized from distantly related bacteria and may serve multiple complex functions that are separate from host mitigation of viral infection. The speculation that groundwater bacteria and archaea are under strong selective pressure to accumulate defense systems, and their viruses to recruit anti-defense systems, isn't supported by experimental evidence.

Second, it is difficult if not impossible for any reviewer to manually inspect these elements and verify their functional assignments: many thousands were detected with genome analysis tools using default settings – these tools often yield false positive results when assigning such specific function for proteins and their cognate genetic features. Moreover, other genetic elements previously associated with defense or viral anti-defense are not mentioned in the main text or supplement, and some systems are closely related in protein homology to elements such as retrons and components of CRISPR. Other mobile elements are known to be quite common in CPR and DPANN and may serve a role in promoting symbiotic interactions with microbial hosts, rather than mitigating viral infection. Some genetic elements may have been missed or incorrectly assigned to better-known categories, due to the automated analyses used by the authors.

Minor points:

Lines 46-47- the estimate of 20-40% is a rather arbitrary and unfounded range, especially with respect to the unknown impact of mortality by infection in groundwater. Also, reference #6 seems to be cited in error here.

Line 69: bacterial communities

Lines 203-206 (and Figure 3 A,B,C): the claim that infection drives expansion (and diversification?) of defense systems is not well supported by the Spearman correlation analysis of virus-host ratios: the Spearman's rho values reported are weak to moderate at best.

Figure 6 (Line 834) Should say “putative mechanism of AcrIIC1 interaction”, as this has not been experimentally validated in the present study.

Reviewer #3

(Remarks to the Author)

1. Overall Summary and Relevance

Summary of the Manuscript The manuscript presents a large-scale metagenomic survey of aquifer microbial communities, aiming to map and characterize their antiviral defense systems (the “defensome”). Using 607 groundwater samples from diverse geological regions in China, the authors reconstruct thousands of metagenome-assembled genomes (MAGs) and identify many known and novel defense genes. They explore the distribution of these defense systems across major prokaryotic lineages (including CPR and DPANN) and discuss ecological and evolutionary patterns, such as trade-offs between antiviral defenses and adaptive traits (e.g., ARGs). They also identify phage-encoded anti-defense mechanisms and highlight a new CRISPR-Cas9–Acr interaction relevant to biotechnology.

Novelty and Significance

Novel Defense Repertoire: Over 190,000 defense genes, with ~90% being novel compared to RefSeq, underscore the aquifer's underestimated reservoir of microbial immune elements. **Ecological Importance:** The manuscript demonstrates how groundwater microbes, particularly certain lineages, harbor dense, diverse antiviral defenses potentially shaped by intense phage infection.

Broad Interest: Results are relevant to microbial ecology, evolutionary biology, and biotechnology (e.g., newly discovered CRISPR-Cas9 variants).

Relevance to Nature Communications: The study's interdisciplinary lens—spanning genomics, ecology, virology, and environmental science—and its focus on a critical freshwater resource (groundwater) align well with the broad readership of the journal.

2. Abstract and Title

Title: Consider refining your title so it incorporates both the scale of your sampling and the central novel findings about aquifer prokaryotic defenses. For instance, highlight that you conducted a large or nationwide survey, underscore the discovery of previously unknown elements in the defensesome, and make clear this work pertains to aquifer ecosystems.

Abstract: It effectively summarizes the goals, methods, and highlights (novel defense genes, phage anti-defense). One improvement is to add a brief concluding sentence on biotechnological or ecological implications of these findings (for example, CRISPR-Cas9 applications). Remove from the abstract the phrase “Compared to the NCBI RefSeq complete prokaryotic genome database” (page 2, lines ~26–27) so that the abstract can focus more on the broader significance rather than on specific comparisons.

3. Introduction and Background

The introduction provides context for phage–microbe coevolution, but some points need refinement. The authors should avoid labeling CPR and DPANN as “keystones” without formal analyses. According to a robust body of literature (Banerjee et al., 2018; Herren & McMahon, 2018; Röttgers & Faust, 2018; and others), identifying keystone taxa requires network analysis, machine learning, functional omics, or experimental validation. If the manuscript retains the term “keystone,” the authors should justify it with evidence or remove the term to avoid misinterpretation.

Have caution when describing these groups as symbionts or widely distributed if the references are not conclusive.

Page 4, line 78: the authors’ statement that CPR and DPANN are symbionts is very strong, so they should verify whether multiple lines of evidence support this lifestyle.

Page 4, line 80: once again, great caution is advised regarding the term “keystone,” and it should be clearly demonstrated if claimed.

Page 4, line 88: they are labeled “abundant and widespread symbionts,” but the manuscript should include multiple references if making such a general assertion.

Readers outside specialized fields might need more detail on why CPR and DPANN are important and how frequently they appear in subsurface environments. Adding more citations on aquifer microbial ecology (Page 2, Lines ~28–33) can strengthen the introduction, and clarifying the role of these lineages at Page 2, ~Line 40 would help readers understand their significance.

4. Methods

Another major concern is the 0.22 μm filter size. Some prior studies used 0.1 μm filters to capture ultra-small prokaryotes (such as CPR). The difference in filter pore size might exclude some smaller cells, and the authors should acknowledge any potential underestimation of CPR/DPANN lineages. This point is highlighted, for instance, by Luef et al. (2015) regarding ultra-small bacterial cells in groundwater.

The sampling plan (607 groundwater samples, 525 new wells) is substantial, covering multiple geological zones, thus supporting large-scale claims. Metagenomic methods (metaWRAP, checkM, dRep, DefenseFinder) are appropriately described, although the authors should justify thresholds such as $\geq 70\%$ completeness and $\leq 10\%$ contamination more explicitly, and discuss the possibility of incomplete recovery of multi-gene defense systems in partially assembled genomes. Please clarify abbreviations like $QS > 70$ (on page 19) and specify precisely what the authors tested via stepwise linear regression (page 22, lines 509–510). In addition, the authors mentioned they conducted all analyses in R (v4.3.1) without providing detailed scripts. Making the R code or an equivalent workflow public would be important for reproducibility.

5. Results

Vast and Novel Defensesome: Approximately 94% of detected defense genes are novel, which indicates a large expansion of microbial immune diversity in groundwater. The supplementary figures (S1–S3) show distributions and correlations with genome size.

Lineage-Specific Defense Patterns: CPR and DPANN present high densities of defense systems, but the authors must be cautious about labeling them as “keystone” if they have not used recognized methods (network analysis, etc.). Large-genome “defense supercarriers” (e.g., certain Pseudomonadota) mirror other environments but also hint at unique aquifer conditions.

Trade-Off with Adaptive Traits: The inverse correlation between defense systems and ARGs, MRGs, or virulence factors is a key ecological finding, pointing toward possible CRISPR-based inhibition of plasmid-borne traits.

Phage Anti-Defense Genes: Observations of anti-CRISPR, anti-RM, anti-Thoeris underlie a complex virus–host arms race in aquifers. Figures S4–S5 (protein structures, Cas9 alignment) reinforce the novelty of these potential interactions.

It was suggested to expand the discussion on whether large genome size correlates with frequent phage encounters (Page 7, Lines ~145–150). Also, consider clarifying the biological significance of functional annotations like COG L, K, and V (lines 281–282). Partial MAGs might underestimate certain defense islands, so the authors could comment on that limitation as well.

6. Discussion and Conclusion

The discussion links intense viral predation with observed patterns of defense density and describes a framework of “cornerstone” vs. “accessory” defense families. Caution is necessary regarding the term “keystone,” since it has a precise meaning in microbial ecology. If Results and Discussion are merged, the text must still maintain a clear logical flow.

I suggest merging the current Results and Discussion into a single combined section, as the Results already contain interpretive material and references. This reorganization could help the manuscript read more fluidly and concisely. Where partial MAGs are concerned, the authors should note how multi-gene defenses such as CBASS might be difficult to detect in fragmented assemblies.

Finally, the conclusion highlights potential biotechnology applications (e.g., novel Cas9 variants) and the relevance for managing antibiotic resistance (via phage–ARG interactions).

7. Language, Structure, and Formatting

I found minor grammatical or typographical issues (e.g., line 30, line 346), so a thorough proofreading is advised.

Figures in the main text can be visually dense; minor modifications (like clarifying labels or using consistent color scales) will improve accessibility.

In the supplementary figures, explicitly linking them to points raised in the text (for instance, “see Fig. S1–S3 for extended distribution analyses”) would help readers navigate the data.

8. Specific Comments Based on Pages and Line Numbers

Page 2, Lines ~28–33: Insert references on aquifer microbial ecology to highlight the uniqueness of this environment.

Page 5, Lines ~90–93: Clearly define your threshold for labeling a defense gene “novel.” For example, is it <80% identity and coverage when compared to RefSeq?

Page 7, Lines ~145–150: Note whether “supercarrier” genomes share certain ecological traits, such as higher metabolic versatility or niche specialization that increases the need for diverse antiviral defenses.

Page 9, Lines ~205–210: Briefly discuss the mechanistic link between CRISPR-based defenses and plasmid-borne ARG acquisition. If the authors have plasmid data in the supplementary material, direct readers there.

Supplementary Figures S4–S5: Including AlphaFold confidence metrics (e.g., pLDDT scores) for protein structure predictions would enhance confidence in the conclusions about novel anti-defense or Cas proteins.

9. Specific Suggestions for Improvement

Explain filter size bias (0.22 μm vs. 0.1 μm) and any underestimation of ultra-small prokaryotes. Use recognized methods or references if calling CPR and DPANN “keystone”; if these methods are not available, remove that term to avoid confusion.

Discuss partial MAG assemblies in the context of multi-gene defense systems. This helps readers interpret possible underestimation of complex systems like CBASS. Merge or reorganize the Results and Discussion if you desire a more streamlined structure, as the current Results section contains numerous interpretive statements. Ensure reproducibility by providing R code or an equivalent workflow in a public repository, clarifying the statistical methods (for example, your stepwise linear regression) and data processing steps. Consider summarizing “cornerstone vs. accessory” defense families in a concise table that describes abundance, possible functions, and typical genomic locations.

10. Overall Recommendation

Recommendation: Major Revisions

Explanation: The study is data-rich and methodologically thorough, presenting a novel view of aquifer microbial immunity. Nonetheless, important clarifications are needed before publication in Nature Communications: (1) Remove or justify the use of “keystone” lineages with formal analyses (e.g., network-based or machine-learning-based identification). (2) Address possible biases in filter size (0.22 μm) and partial MAGs, which may exclude key taxa or omit multi-gene defenses. (3) Improve clarity and structure, for example by merging the Results and Discussion sections if that would reduce redundancy. (4) Strengthen reproducibility by providing code and clarifying all thresholds (especially for “novelty” definitions and statistical methods).

With these revisions in place, the manuscript will provide a significant contribution to our understanding of viral-host interactions in subsurface ecosystems and will be well-suited for publication in Nature Communications.

Version 1:

Reviewer comments:

Reviewer #1

(Remarks to the Author)

The authors addressed all my questions / suggestions.

Congratulations for the work.

Reviewer #2

(Remarks to the Author)

The authors have addressed most of my concerns in their revised manuscript and they have made substantial improvements to their study. I was especially pleased to see the efforts to experimentally validate several defense systems via recombinant phage infection (plaque) assays in *E. coli*. However, a remaining weakness of the study is the lack of parallel omics data to show which systems are actively deployed in situ (this was not directly addressed in the authors' response). I imagine such datasets as metaproteomes or metatranscriptomes may not be available from the original sampling effort, and as such, couldn't be provided for a revision of this study.

Reviewer #3

(Remarks to the Author)

Thank you for the thorough revision. You have addressed all substantive issues raised in the first round. In particular, the “keystone” terminology has been removed or appropriately qualified; the potential 0.22 μm -filter bias and limits of partial MAG assemblies are now discussed; sequence-novelty thresholds, AlphaFold confidence metrics and every software parameter are clearly stated; the full R/Shell workflow is openly available; and you added plaque-assay validation for three representative defence systems (R-M IIG, SoFIC and CBASS), which substantiates the in-silico predictions. The Results and Discussion sections have been merged, improving flow and eliminating redundancy, and language and figure legends

have been polished.

Only very small editorial items remain (for example, add units to the y-axis of Supplementary Fig. S6 panel D; provide a short Materials-and-Methods subsection specifically for the plaque assays; insert hyphens in "R-M IIG" where needed). These can be handled easily during production.

With those minor tweaks, the manuscript is ready for publication in Nature Communications.

Response to the Reviewers' comments

Response to the Reviewer #1

General Comment: Li and co-workers propose a large-scale analysis of the groundwater prokaryotic anti-phage defensome. This is a timely work that follows the trend of a growing number of studies focusing on the environmental defensome. I have a few comments/suggestions that I hope the authors will find useful.

Response: Thank you very much for your confirmation on our work. Following your comments and suggestions, we have seriously revised the manuscript to address all the concerns and editorial requirements. Please find below with item-by-item responses to the specific comments/suggestions raised by the Reviewer #1.

Comment No.1: The recent use of machine learning models applied to the prediction of novel defense systems (<https://www.biorxiv.org/content/10.1101/2025.01.08.631726v1>) shows significant improvement over DefenseFinder concerning the number of novel families detected. Maybe it would be appropriate to see if more recent tools provide a more fine-grained view of the defensome and shift any of the conclusions?

Response: Thank you very much. DefensePredictor¹ is a timely tool that will greatly advance the relevant studies in this field. Following the reviewer's suggestion, we applied DefensePredictor on the high-quality MAG set (1,626 MAGs) in the present study, and found it could identify more defense genes than DefenseFinder (Fig. S6A), the results obtained using both software are consistent regarding the identified number of defense genes (Fig. S6B). DefensePredictor identified a larger range of variation in the number of defense genes than DefenseFinder (Fig. S6C), covering 78.4% defense genes predicted by DefenseFinder (Fig. S6D). This is not surprising because DefensePredictor uses DefenseFinder results as its positive training set¹. We added a

brief discussion about the DefensePredictor results in Lines 141-145. This enables to give a more comprehensive view of the huge potential of digging new defense genes in groundwater. The consistency of the results from the above two software (Fig. S6B) also suggests the validity of using DefenseFinder in this study. The revised contents (Lines 141-145 in the new version) read:

“Alternatively, DefensePredictor, a recently developed tool that utilizes protein language models to identify defense genes⁴², is used on our 1,626 high-quality MAGs (DefensePredictor results added in Fig. S6), we find more candidate defensive genes (2.4 times than DefenseFinder, Fig. S6A), which further highlights the huge potential of digging new defense genes in groundwater.”

Fig. S6. DefensePredictor results on 1,626 high-quality MAGs. (A) The number of defense genes identified by DefenseFinder and DefensePredictor on 1,626 high-quality MAGs. (B) Results derived from the two software on each MAG. The MAGs were sorted according to the DefenseFinder results from the smallest to the largest. (C) Density plots showing number proportion of defense genes identified using the two software among MAGs where defense genes were detected. Dashed lines represent the number of defense genes corresponding to the maximum density. (D) The proportion

of co-identified defense genes (by both software) and those identified with DefenseFinder or DefensePredictor.

Although DefensePredictor identifies more defense genes than DefenseFinder, it hardly provides classified information which is essential for subsequent analyses. DefensePredictor, designed by utilizing protein language model, aims to identify as many defensive genes as possible, while DefenseFinder uses predefined rules (e.g., mandatory or accessory genes, gene organization) to determine which defense genes to retain and assigns them known classifications. DefensePredictor is better suited for discovering entirely new defense genes in specific microorganisms, as demonstrated in their work on 69 *E. coli* strains¹, while DefenseFinder results seem more suitable for investigating environmental “defensome” in the present study. Considering another reviewer mentioned the potential false positives, we take the caution to add a discussion on the DefensePredictor results in the main text and give more details (Fig. S6) in Supplementary Information at this stage. In future work, we would like to take advantages of DefensePredictor for the purpose of unveiling the unknown immune mechanisms in groundwater microorganisms.

Comment No.2: The current know-how on phage anti-defense systems is still very limited, and the majority of DefenseFinder HMMs have been obtained for the major anti-defense systems (anti-RM, anti-CRISPR, etc), while non-first-line systems’ HMMs are still poorly accurate or not built. Your 1.6% value (625 in 40,302 phages) demonstrates how far you are to have a solid view on the panoply of anti-defense systems. I would either remove the section on anti-defense, or caution the reader that this section risks to be highly inaccurate and poorly representative of the actual anti-defense landscape.

Response: Thank you very much. We have added a discussion to emphasize the limitations of the anti-defense results, acknowledging the gap in understanding the anti-

defense landscape, cautioning readers the limitations of these findings, and providing the insights. The revised contents read:

Lines 371-374: “To explore phage anti-defense mechanisms in groundwater, we identified 712 anti-defense genes across nine distinct anti-defense types from 625 phages, accounting for only 1.6% of all 40,302 phages extracted from 27,578 bacterial MAGs (see Fig. S18 for the distribution of anti-defense genes).

Lines 529-533: “Note that DefenseFinder HMMs are primarily developed for major anti-defense systems (e.g., anti-RM, anti-CRISPR)¹¹⁰, and non-first-line systems either lack reliable HMMs or exhibit poor accuracy, it is far from sufficient for understanding anti-defense landscape based on the predicted results because of the limitation of current knowledge on phage anti-defense systems.”

Comment No.3: Did you observe any effect of groundwater sampling depth (until 600 m according to your previous study) on the defensome composition?

Response: Many thanks. We conducted additional analyses to assess the effects of sampling depths, well types, and geographical contexts on the microbial defensome. As results, no significant correlation was found between the number/density of defense systems and sampling depth (Fig. S8A, B). However, the MAGs from reconstructed wells exhibit slightly higher defense density than in newly constructed wells (Fig. S8C). We further compared the defense density across seven geographic zones (Fig. S8E), and found the highest in zones I (Northeast Plain-Mountain) and VII (Qinghai-Tibet plateau Alpine frozen soil), and the lowest in zones III (South China bedrock low mountain foothill) and VI (Northwest arid desert), largely attributed to the distinct geo-environmental conditions. We added the relevant information in the revised version, which reads (Lines 168-175):

“Metagenomic analysis reveals a slightly higher defense density in reconstructed wells than in newly constructed wells (Fig. S8C). We further compared the defense density

across seven geographic zones (Fig. S8E), and found the highest in zones I (Northeast Plain-Mountain) and VII (Qinghai-Tibet plateau Alpine frozen soil), and the lowest in zones III (South China bedrock low mountain foothill) and VI (Northwest arid desert), largely attributed to the distinct geo-environmental conditions. While the underlying mechanisms remain unclear, these findings suggest the complex effects of ecological contexts on the composition of microbial community and their defensome.”

Fig. S8. Effects of sampling depths, well types, and geographical contexts on the microbial defensome. (A) Correlation between defense system (DS) number and burial depth of groundwater. (B) Correlation between defense system density (per MAG per Mb) and burial depth of groundwater. (C) Comparison of defense system density (per MAG per Mb) among newly constructed confined wells (NC), newly constructed phreatic wells (NP), and reconstructed wells (R). (D) Comparison of defense system density (per MAG per Mb) in confined water and phreatic water. (E) Comparison of defense system density (per MAG per Mb) across seven geographic zones. Detailed zone annotations are provided in Fig. S2A. Significant differences were assessed using the Wilcox test with Bonferroni-adjusted P values. “*” “***”, “****”, “*****” indicates $P < 0.05, 0.01, 0.001, 0.0001$.

Comment No.4: We have now accumulated evidence of the role of these defense systems against multiple classes of MGEs (and not only phages). Maybe change the term “antiphage” to “anti-MGE” across the text?

Response: Thank you very much. We have replaced “antiphage” with “anti-MGE” throughout the manuscript.

Comment No.5: Fig. 5C Are these prophages? If not, the authors should disentangle between prophages and non-prophages. Does the ‘chromosome’ box plot contain MGEs? If so, they should control for this, by removing chromosomal MGEs.

Response: Thank you very much. We disentangled the results between phages and prophages (Fig. 5A-D). The ‘chromosome’ box plot doesn’t include major MGEs because we have excluded phages, prophages, plasmids, ICEs/IMEs, and integrons from it.

Fig. 5. Mobile genetic elements (MGEs) as carriers of accessory defense systems and their defense island reservoir. (A) Distribution of bacterial phyla of MGE hosts. (B) Relative percentages of MGEs across phyla, showing only phyla containing ≥ 500 MGEs. (C) Comparison of defense system (DS) density (per kb) between chromosome and other MGEs (Wilcoxon test, Bonferroni-adjusted P values, “*****” represents adjusted $P < 0.0001$). (D) Heatmap of observed/expected (O/E) ratios for colocalization of DSs and MGEs. Expected values were calculated by multiplying the total number of systems in the given defense family by the fraction assigned to chromosome, or the other MGEs. (E) Number of defense systems across different defense families, with families having < 50 systems grouped as “Others”. (F), (G), (H) Comparison of defense genes (DGs) per defense island (DI), defense island length (genes), and proportion of defense genes within defense islands across CPR, Pseudomonadota, and other phyla (Wilcoxon test, Bonferroni-adjusted P values). Number of DIs analyzed are shown as n values. (I) Heatmap of O/E ratios for colocalization of defense systems and DIs. Expected values were calculated by multiplying the total number of systems in a defense family by the fraction assigned to DIs or non-DIs.

Comment No.6: What about ICEs/IMEs? Integrons? These are really important for the analyses.

Response: Thank you very much. In previous analyses, we focused on phages and plasmids as they are the dominant MGEs mostly interested to the community. In the revised version, we incorporated the results of ICEs/IMEs and integrons, which are main integrative elements in microbial chromosomes (Fig. 5A-D). Consistent with previous studies², we also observed higher colocalization of defense genes with ICEs/IMEs and integrons. These results have been added in the revised version (Lines 302-319), which reads:

“We also investigated the roles of major families of MGEs (plasmids, phages, prophages, integrons, and ICEs/IMEs) in the mobility of defense systems⁷³. Previous studies in other ecosystems have shown that MGEs can carry defense systems, promoting their spread for selfish propagation⁴⁷. In groundwater ecosystems, we identified 163,897 plasmids, 40,302 phages/prophages, 2,697 integrons, and 606 ICEs/IMEs across 27,578 bacterial MAGs, with the highest representation in Pseudomonadota (Fig. 5A, see Fig. S15 for detailed distribution of MGEs). Notably,

CPR exhibits the highest proportion of phages/prophages (73% of their MGEs, Fig. 5B), aligning with the finding that they are under intense viral infections³⁷. Defense systems are more densely localized on MGEs compared to chromosomal regions (Fig. 5C), with all types of MGEs carrying a higher-than-expected number of defense systems from a wide range of defense families (Fig. 5D). Unlike first-line defense systems such as R-M, SoFIC, CRISPR-Cas, and MazEF, many accessory immune systems are preferentially positioned on MGEs, suggesting that microorganisms dynamically adapt to their environments by horizontally acquiring or discarding these accessory systems⁶. Additionally, distinct localization preferences among defense families are evident, with systems like BstA more commonly found on phages/prophages, while dGTPase tends to localize on plasmids (Fig. 5D). These findings highlight the complex interplay between defense systems and MGEs, with specific families favoring certain MGEs for their dissemination and maintenance⁵².”

Comment No.7: Fig.1 caption: “from a previous study”. Which one?

Response: Many thanks. Citation marks are added in the caption. The data are from the literature: “Beavogui, A. *et al.* The defensome of complex bacterial communities. *Nat. Commun.* **15**, 2146 (2024).”

Comment No.8: Page7: line157. This sentence comes in contrast to dozens of papers in the literature that support HGT as a main source of shuttling and accumulation of defense systems (notably in Dis). Maybe clarify?

Response: Thank you very much. We fully agree that HGT is the primary mechanism for the shuttling and accumulation of defense systems. Upon re-evaluation, we found that the weak negative correlation between phylogenetic depth and the number of defense systems lacks statistical power and may lead to confusion. To ensure clarity, we have removed the content related to phylogenetic depth from the Results.

Comment No.9: The authors should caution for the fact that a “low defense system density” (or vice-versa) in a particular clade (e.g., CPR, DPANN) might arise simply by the fact that such clades harbor more exotic or unknown DSs not covered by the existing detection methods (or inversely, are very well covered by the existing methods).

Response: Many thanks. Yes, we fully agree. The rapid discovery of new DSs highlights that there is still much to learn about microbial immunity, particularly in underexplored clades. To address this, we have included a cautionary discussion in the manuscript (Lines 231-234) to avoid confusion.

“It is worthy of noted that the observed ‘low defense system density’ in certain clades (e.g., CPR, DPANN) might also reflect the presence of exotic or unknown defense systems not currently detectable by existing methods.”

Comment No.10: Page 10: section adaptive traits. Wouldn't this inverse trend be already expected? If one has more defense systems, in theory, you should curb HGT, and therefore the acquisition of adaptive traits as the ones mentioned. Such “intriguing” finding as mentioned by the authors, seems actually expected, and was already described in the literature: <https://academic.oup.com/nar/article/51/9/4385/7132345>

Response: Thank you very much. We cited this article in our previous manuscript (Reference No.71). The cited article focused on the ICEs/IMEs, indicating the correlation between DSs, ARGs, and VFGs, while our study extends to the whole prokaryotic genomes, particularly in groundwater ecosystems where the trade-off between DSs and ARGs would be of significance for controlling environmental ARG risks. We revised the related contents as such:

Lines 270-275: “In integrative and conjugative/mobilizable elements (ICEs/IMEs)⁶⁵, previous study has indicated the trade-offs between defense systems and antibiotic resistance genes (ARGs). Here, we find that such trade-offs can be extended to the whole prokaryotic genomes, evidenced by the inverse relations between the number of

defense systems and the acquisition of adaptive traits such as ARGs, heavy-metal resistance genes (MRGs), and virulence factor genes (VFGs) in microorganisms of groundwater environment (Fig. 4A).”

Lines 290-300: “Here, the underlying trade-off mechanisms could help to interpret the corresponding characteristics of microbial adaptive traits under varying virus-host interactions. For example, a recent study observed the natural water “self-purification” phenomenon⁶⁹, where pathogens carrying abundant ARGs are suppressed by phages. This could be partially explained by trade-offs between defense systems and ARGs. As ARGs continually pose a significant challenge to human and environmental health, phage therapy has emerged as a promising alternative to combat antibiotic-resistant bacterial infections⁷⁰⁻⁷². In this regard, the trade-offs between defense systems and adaptive traits strengthen the potential of using phages to inhibit super-resistant bacteria, because these bacteria often possess limited anti-MGE capabilities, rendering them more susceptible to phage-mediated elimination²¹.”

Comment No.11: Avoid acronyms in the abstract (e.g., CPR). Also use “R-M” instead of “RM” to agree with the official notation.

Response: Thank you. We have fixed the acronyms problem in abstract. We also revised the manuscript to use “R-M” instead of “RM”, aligning with the official notation.

Comment No.12: It would be fair if the authors could acknowledge the work that first introduced the term defensome applied to antiphage defenses (<https://www.nature.com/articles/s41467-024-46489-0>).

Response: Thank you very much. We cited the work by Beavogui, A. *et al.* in our previous manuscript. In the revised version, we stressed their contribution in this area (Lines 65-68), which reads:

“Beavogui, A. *et al.* (2024) first proposed the term “defensome” and reported that bacterial communities exhibit diverse defense strategies against phages across a wide range of uncultured microbes in soil, marine, and human gut systems²².”

Comment No.13: Page 5: Please avoid using DefenseFinder family nomenclature with underscores.

Response: Many thanks. Fixed.

Comment No.14: Please avoid the term “cornerstone” defense systems. For the sake of uniformity, I would suggest the term already in use in the literature “first-line” defense systems.

Response: Thank you very much. We used the term “first-line” across the manuscript as suggested.

Comment No.15: Please provide the wrapper scripts for all the analyses. These should be made public for questions of clarity.

Response: Thank you very much. We have submitted all wrapper Shell scripts and R codes used in this study in GitHub (<https://github.com/lianmsu/aquifer-defensome>).

Lastly, we would like to express our sincere thanks to the anonymous reviewer. The comments and suggestions raised by the reviewer are of great significance to the improvement of our work.

Response to the Reviewer #2

General comment: Li and coauthors carried out a robust survey of bacterial and archaeal defense, and viral anti-defense systems in genomes reconstructed from the groundwater biosphere. The manuscript describes a systematic effort to generate a compendium of genetic elements that have potential function in defense for their hosts, or separately, for viral resistance to host defenses. However, I find two major issues in review of the manuscript. First, the study lacks additional omics or experimental data (metatranscriptomes, proteomes, or genetic experiments) that could demonstrate that at least some of these defense systems are expressed, or even function as predicted. These defense systems are best characterized from distantly related bacteria and may serve multiple complex functions that are separate from host mitigation of viral infection. The speculation that groundwater bacteria and archaea are under strong selective pressure to accumulate defense systems, and their viruses to recruit anti-defense systems, isn't supported by experimental evidence.

Second, it is difficult if not impossible for any reviewer to manually inspect these elements and verify their functional assignments: many thousands were detected with genome analysis tools using default settings – these tools often yield false positive results when assigning such specific function for proteins and their cognate genetic features. Moreover, other genetic elements previously associated with defense or viral anti-defense are not mentioned in the main text or supplement, and some systems are closely related in protein homology to elements such as retrons and components of CRISPR. Other mobile elements are known to be quite common in CPR and DPANN and may serve a role in promoting symbiotic interactions with microbial hosts, rather than mitigating viral infection. Some genetic elements may have been missed or incorrectly assigned to better-known categories, due to the automated analyses used by the authors.

Response: Thank you very much. Following the reviewer's comments and suggestions, we have done our best by conducting additional genetic experiments to validate the anti-phage function of representative defense systems identified in groundwater. For the sake of caution, we invited our peers from the school of life sciences working together with us to cross-validate and guarantee the high quality of the experiments. Meanwhile, we employed DefenseFinder (a widely used tool)³⁻⁷ to identify defense systems and to filter the results for ensuring low bias through predefined HMM profiles and rules⁸, retaining only 16.1% (of an initial pool of 1,183,670 homologs of defense genes) for downstream analyses according to these rules. While the present study revealed a positive correlation between prokaryotic defense systems and phage infection intensity in groundwater, with similar evidences from other ecosystems in recent studies^{7,9}, the weak to moderate Spearman's rho values still require future validation from omics or experimental evidences. As far as the false positive issue, DefenseFinder applies well-defined conservative rules for each defense system to retain only the HMM hits that satisfy the genetic architecture of the system, these decision rules are typically defined by a list of mandatory, accessory, or forbidden proteins necessary for the detection of a given system, along with the corresponding genetic architecture. We also re-examined and clarified the parameters for all bioinformatic tools used in this study.

To address the concerns from the Reviewer #2, we provide a detailed response as well as corresponding revisions as follows:

(1) Additional genetic experiments to validate function of the representative defense systems identified.

Using plaque assays, we found that the identified defense systems are functional against phage infection. Importantly, we selected both the close and distant homologs (randomly selected from the bottom 50% of results, Fig. S9) for validation, supporting the inference that most identified defense genes function as predicted. In the revised

manuscript, we give a full description of the additional experiments (see Methods, Results and Discussion), which reads:

Lines 491-516: R-M and SoFIC systems were identified as the most abundant defense systems in groundwater prokaryotes, representing first-line immunity (Fig. 1A), with R-M IIG being the most prevalent R-M subtype. Therefore, R-M IIG and SoFIC were selected for experimental validation of their anti-phage activity using plaque assays. In addition, the CBASS system, representing accessory immunity, was also selected for validation. For R-M IIG and SoFIC, two gene variants were chosen for each system: one closely related to the corresponding HMM profile (i.e., the top homology candidates prioritized by hit score) and one more distantly related (randomly selected from low-homology candidates). Distant homologs were selected using the following median-based thresholds: R-M IIG—hit score <668, hit_profile_cov <0.94, hit_seq_cov <0.94, hit_seq_len <1044; SoFIC—hit score <375, hit_profile_cov <0.96, hit_seq_cov <0.95, hit_seq_len <373). In total, five defense systems were evaluated (see Supplementary Data 10 for detailed information). The coding sequences of the selected genes were commercially synthesized in GenScript and cloned into either the pQE-60 (R-M IIG and SoFIC) or pETDuet-1 (CBASS) expression vector. Recombinant plasmids were transformed into *Escherichia coli* B (BL21), with empty vector controls included for comparison. Plaque assays were conducted following established protocols¹⁰⁶. A single colony from a fresh LB agar plate was inoculated into LB broth containing ampicillin (50 µg/mL) and grown at 37°C until reaching an OD600 of ~0.6. For pETDuet-1 vector, protein expression was induced with 0.2 mM isopropyl β-D-1-thiogalactopyranoside (IPTG). After an additional hour of incubation at 16°C, 1.5 mL of the bacterial culture was mixed with 12.5 mL of 0.6% LB top agar containing ampicillin (50 µg/mL) and IPTG (0.2 mM, for pETDuet-1 vector only), and the mixture was poured onto LB plates containing ampicillin (50 µg/mL). Each plate was then spotted with 5 µL aliquots of serially diluted phage suspensions (10⁶ to 10² for T1 and T7 phages). For statistical comparison, the plaque forming unit (PFU) was counted at

7 h post-infection. Plates were incubated at 25°C overnight before imaging. All experiments were performed in three batches for replicability.”

Lines 180-184: “Further, to partially address the limitations due to the lack of multi-omics data from the 607 groundwater samples, we conducted genetic experiments to validate the functions of the representative defense systems against phage infection as predicted (see Fig. S9 for plaque assays). Future studies incorporating multi-omics approaches and functional experiments are essential to further validate and expand upon these findings.”

Fig. S9. Plaque assays showing phage infection of *Escherichia coli* B (BL21) transformed with plasmids carrying representative defense systems or empty vector

(control). Ten-fold serial dilutions of phages (T1 and T7) were spotted onto bacterial lawns. **(A)** Plaque assay image after incubation at 25°C overnight. **(B)** Viral plaque forming unit (PFU) measured by plaque assay at 7 h post-infection. n = 3 biological replicates. Statistic differences between empty vector and vectors carrying defense systems were assessed using paired t test. “*”, “*****” indicates P < 0.05, 0.0001.

(2) The complex functions of defense genes.

We acknowledge that some defense genes may have multiple complex functions beyond mitigating viral infection. However, we used DefenseFinder, a tool employing predefined HMM profiles and rules (a list of mandatory, accessory, or forbidden proteins necessary for the detection of a given system, along with the corresponding genetic architecture), could help to filter results and ensure low bias⁸. In this study, only 16.1% (from an initial pool of 1,183,670 homologs of defense genes) was retained for downstream analyses according the above rules. Since this tool has been widely used in previous studies³⁻⁷ to identify defense genes in other systems, it is acceptable to be used in groundwater despite some limitations, as discussed in our revised manuscript (Results and Discussion), which reads:

Lines 176-180: “Some defense genes might have multiple complex functions beyond mitigating viral infection, while the DefenseFinder, a tool employing predefined HMM profiles and rules (a list of mandatory, accessory, or forbidden proteins necessary for the detection of a given system, along with the corresponding genetic architecture), could help to filter results and ensure low bias¹⁸.”

(3) The speculation’s further evidence.

The reviewer also raised a very interesting question for us to consider in the future: to provide experimental evidence for the speculation that “groundwater bacteria and archaea are under strong selective pressure to accumulate defense systems, and their viruses to recruit anti-defense systems.” Since our study focused on the defensome of environmental microorganisms, we prefer to discuss this issue briefly with some evidences from recent other studies (e.g. Gaborieau, B. *et al.*⁷, Zhang, Q. *et al.*⁹) at this

stage and indicate the limitations in the revised manuscript (Lines 250-256), which reads:

“While the weak to moderate Spearman’s rho values necessitate the need for additional omics or experimental evidence to substantiate this hypothesis, recent studies provide complementary support. Experimental evidence indicates that the number of defense systems in a bacterial strain could affect the level of susceptibility to already infecting phages⁶³. And a significant positive correlation ($r = 0.70$, $P < 0.0001$) between defense system abundance and phage abundance was observed in activated sludge system⁶⁴.”

(4) About the false positives.

We re-examined and clarified the parameters for all bioinformatic tools used in this study. Additionally, we used both DefenseFinder⁸ and DefensePredictor¹ to cross-validate the reliability of identified defense systems, and reached consistent results. Our analysis revealed that approximately 78.4% of the DefenseFinder results were corroborated by DefensePredictor when using a stringent probability cutoff of >0.999 , further supporting the robustness of the results.

Considering the advantages with less false positives, we used DefenseFinder as a major tool by employing individual gathering threshold (GA) parameters for each HMM profile, which is similar to the functional annotation tool KofamScan, serving as the first-line guarantee of results’ reliability. Since many defense systems require synergistic actions of multiple genes, DefenseFinder applies well-defined rules for each defense system to retain only the HMM hits that satisfy the genetic architecture of the system. The decision rules are typically defined by a list of mandatory, accessory, or forbidden proteins necessary for the detection of a given system, along with the corresponding genetic architecture. As noted by authors of DefenseFinder⁸, “The current settings of DefenseFinder are optimized for a conservative detection, as only full systems are present.” In this study, from an initial pool of 1,183,670 potential defense gene homologs, only 16.1% defense genes were assigned as part of complete systems and

retained as final results. While false positives are inevitable, the defense systems identified using these cutoffs and rules are reliable and acceptable for capturing an overview of the environmental defensesome. Similarly, other bioinformatic tools used in this study employ either strict default settings or parameters widely accepted in the field. This study represents a comprehensive census of the antiviral arsenal of groundwater prokaryotes. This census may change in its details as many antiviral systems probably remain to be discovered and software parameters are refined, but the general trends observed in this study are expected to remain largely consistent.

Below is an example from the work by Tesson, F. *et al.* (2022), illustrating how selecting an appropriate hit score can maintain a high true positive rate while minimizing false positives⁸. Additionally, applying rules for complete systems can further reduce false positives.

Response figure 1, cited from Tesson, F. *et al.*⁸ a. Analysis of true and false positives for Lamassu's proteins depending on the hit score. True positive rates were computed as DefenseFinder detection on protein from (Doron et al 2018) divided by the number of proteins detected by Doron and colleagues. False positives were calculated by running DefenseFinder on genomes where Doron's and colleagues have not found any system. Normalized false positive rate is computed as $FP/MaxFP$, where False positive (FP) is the number of off-target proteins at each GA cut and Maximum false positive (MaxFP) is the number of off-target proteins without any GA cut. The continuous line counts every false hit protein whereas discontinuous lines represent false positives that are present with other proteins of the system. The vertical bar represents the GA cut that was chosen for DefenseFinder.

In the revised manuscript, we provided additional information (Fig. S6) about the issues and limitations as follows:

Lines 235-242: “This study provides a comprehensive assessment of the defensome of prokaryotes in groundwater. While the well-established bioinformatic tools (annotation methods with stringent filtering criteria)¹⁸ are employed with carefully selected parameters to ensure the reliability of results, some degree of false positives or misclassification could still not be avoided^{18,23}. The evolving nature of defense system discovery^{1,8,13,16,17} and refinement of detection tools⁴² may also lead to updates in specific annotations over time. Nevertheless, we expect the general trends observed in this study will remain robust, offering valuable insights into the defensome of environmental microbes.”

Fig. S6. DefensePredictor results on 1,626 high-quality MAGs. (A) The number of defense genes identified by DefenseFinder and DefensePredictor on 1,626 high-quality MAGs. (B) Results derived from the two software on each MAG. The MAGs were sorted according to the DefenseFinder results from the smallest to the largest. (C) Density plots showing number proportion of defense genes identified using the two software among MAGs where defense genes were detected. Dashed lines represent the number of defense genes corresponding to the maximum density. (D) The proportion

of co-identified defense genes (by both software) and those identified with DefenseFinder or DefensePredictor.

(5) Additional information about other mobile genetic elements.

In response to the reviewer's comments, we provided additional results about other mobile genetic elements (MGEs) associated with defense or viral anti-defense, including ICEs/IMEs and integrons, which are main integrative elements in microbial chromosomes (Fig. 5A-D, Fig. S14).

The reviewer also indicated that “some systems are closely related in protein homology to elements such as retrons and components of CRISPR.” In the HMM outputs of DefenseFinder, there may indeed be instances where a protein is assigned to different type of defense genes. To address this concern, we take advantages of the DefenseFinder with the predefined rules for system completeness, which can help to resolve such ambiguities and eliminate overlapping assignments of the final results.

In general, HGT involves three main types of MGEs: plasmids, phages, and integrative elements. Plasmids are typically extrachromosomal, while phages can exist either extrachromosomally or as dormant prophages within the host genome. Integrative elements, on the other hand, are stably integrated into the bacterial chromosome. Among these, autonomous elements are a subset of integrative elements that can excise themselves from the chromosome and reinsert elsewhere, facilitated by enzymes such as transposases or integrases/excisionases (found in ICEs and IMEs). In our previous analyses, we focused on phages and plasmids as they are the most dominant MGEs and of primary interest to the community. Our additional findings are consistent with those from previous studies², both showing higher carriage of defense genes by ICEs/IMEs and integrons (Fig. 5A-D, Fig. S14). The revised paragraph (Lines 302-319) reads:

“We also investigated the roles of major families of MGEs (plasmids, phages, prophages, integrons, and ICEs/IMEs) in the mobility of defense systems⁷³. Previous studies in other ecosystems have shown that MGEs can carry defense systems,

promoting their spread for selfish propagation⁴⁷. In groundwater ecosystems, we identified 163,897 plasmids, 40,302 phages/prophages, 2,697 integrons, and 606 ICEs/IMEs across 27,578 bacterial MAGs, with the highest representation in Pseudomonadota (Fig. 5A, see Fig. S15 for detailed distribution of MGEs). Notably, CPR exhibits the highest proportion of phages/prophages (73% of their MGEs, Fig. 5B), aligning with the finding that they are under intense viral infections³⁷. Defense systems are more densely localized on MGEs compared to chromosomal regions (Fig. 5C), with all types of MGEs carrying a higher-than-expected number of defense systems from a wide range of defense families (Fig. 5D). Unlike first-line defense systems such as R-M, SoFIC, CRISPR-Cas, and MazEF, many accessory immune systems are preferentially positioned on MGEs, suggesting that microorganisms dynamically adapt to their environments by horizontally acquiring or discarding these accessory systems⁶. Additionally, distinct localization preferences among defense families are evident, with systems like BstA more commonly found on phages/prophages, while dGTPase tends to localize on plasmids (Fig. 5D). These findings highlight the complex interplay between defense systems and MGEs, with specific families favoring certain MGEs for their dissemination and maintenance⁵².”

Fig. 5. Mobile genetic elements (MGEs) as carriers of accessory defense systems and their defense island reservoir. (A) Distribution of bacterial phyla of MGE hosts. **(B)** Relative percentages of MGEs across phyla, showing only phyla containing ≥ 500 MGEs. **(C)** Comparison of defense system (DS) density (per kb) between chromosome and other MGEs (Wilcox test, Bonferroni-adjusted P values, “****” represents adjusted $P < 0.0001$). **(D)** Heatmap of observed/expected (O/E) ratios for colocalization of DSs and MGEs. Expected values were calculated by multiplying the total number of systems in the given defense family by the fraction assigned to chromosome, or the other MGEs. **(E)** Number of defense systems across different defense families, with families having < 50 systems grouped as “Others”. **(F), (G), (H)** Comparison of defense genes (DGs) per defense island (DI), defense island length (genes), and proportion of defense genes within defense islands across CPR, Pseudomonadota, and other phyla (Wilcox test, Bonferroni-adjusted P values). Number of DIs analyzed are shown as n values. **(I)**

Heatmap of O/E ratios for colocalization of defense systems and DIs. Expected values were calculated by multiplying the total number of systems in a defense family by the fraction assigned to DIs or non-DIs.

Fig. S15. Mobile genetic elements (MGEs) identified from bacterial genomes. For ease of visualization, phyla with fewer instances of specific MGEs were grouped into the category “Other phyla”.

While MGEs play roles in host’s defense, the reviewer indicate that they are common in CPR and DPANN and may promote symbiotic interactions with microbial hosts. Indeed, MGEs often act as important reservoirs of adaptive functions, contributing to various host traits such as resistance genes, virulence factors, and auxiliary metabolic genes. In the revised manuscript, we provided additional results about MGEs to show their importance on host defense. However, we don’t mean MGEs are exclusively dedicated to host defense, they surely serve diverse biological roles beyond mitigating viral infection.

Minor points:

Lines 46-47- the estimate of 20-40% is a rather arbitrary and unfounded range, especially with respect to the unknown impact of mortality by infection in groundwater. Also, reference #6 seems to be cited in error here.

Response: Thank you for pointing this out. We have revised the Introduction section to ensure accuracy (Lines 41-43). Meanwhile, the error in citation of reference #6 has been corrected.

“Phage infection is a major driver of prokaryotic evolution, accounting for about 20% of daily bacterial mortality, with more than 10^{23} infections occurring per second in the oceans alone⁵.”

Line 69: bacterial communities

Response: Thank you. Corrected (Line 66).

Lines 203-206 (and Figure 3 A, B, C): the claim that infection drives expansion (and diversification?) of defense systems is not well supported by the Spearman correlation analysis of virus-host ratios: the Spearman’s rho values reported are weak to moderate at best.

Response: Thank you very much. Recently, Gaborieau, B. *et al.* found that the number of defense systems in a bacterial strain could affect the level of susceptibility to already infecting phages through experiments⁷. Zhang, Q. *et al.* reported a significant positive correlation ($r = 0.70$, $P < 0.0001$) between defense system abundance and phage abundance in activated sludge system⁹. Since our study focused on the defensome of environmental microorganisms, we prefer to discuss this issue briefly with some evidences from recent other studies (e.g. Gaborieau, B. *et al.*⁷, Zhang, Q. *et al.*⁹) at this stage and indicate the limitations in the revised manuscript (Lines 250-256), which reads:

“While the weak to moderate Spearman’s rho values necessitate the need for additional omics or experimental evidence to substantiate this hypothesis, recent studies provide complementary support. Experimental evidence indicates that the number of defense systems in a bacterial strain could affect the level of susceptibility to already infecting phages⁶³. And a significant positive correlation ($r = 0.70$, $P < 0.0001$) between defense system abundance and phage abundance was observed in activated sludge system⁶⁴.”

Figure 6 (Line 834) Should say “putative mechanism of AcrIIC1 interaction”, as this has not been experimentally validated in the present study.

Response: Many thanks. Changed as suggested (Line 933).

Again, we would like to express our gratefulness to the anonymous reviewer for raising valuable comments and suggestions, which are indeed of great help to the improvement of this manuscript.

Response to the Reviewer #3

Comment No.1 Overall Summary and Relevance

Summary of the Manuscript: The manuscript presents a large-scale metagenomic survey of aquifer microbial communities, aiming to map and characterize their antiviral defense systems (the “defensome”). Using 607 groundwater samples from diverse geological regions in China, the authors reconstruct thousands of metagenome-assembled genomes (MAGs) and identify many known and novel defense genes. They explore the distribution of these defense systems across major prokaryotic lineages (including CPR and DPANN) and discuss ecological and evolutionary patterns, such as trade-offs between antiviral defenses and adaptive traits (e.g., ARGs). They also identify phage-encoded anti-defense mechanisms and highlight a new CRISPR-Cas9–Acr interaction relevant to biotechnology.

Novelty and Significance

Novel Defense Repertoire: Over 190,000 defense genes, with ~90% being novel compared to RefSeq, underscore the aquifer’s underestimated reservoir of microbial immune elements. Ecological Importance: The manuscript demonstrates how groundwater microbes, particularly certain lineages, harbor dense, diverse antiviral defenses potentially shaped by intense phage infection.

Broad Interest: Results are relevant to microbial ecology, evolutionary biology, and biotechnology (e.g., newly discovered CRISPR-Cas9 variants).

Relevance to Nature Communications: The study’s interdisciplinary lens—spanning genomics, ecology, virology, and environmental science—and its focus on a critical freshwater resource (groundwater) align well with the broad readership of the journal.

Response: Thank you very much for your summary and confirmation of our work (Novel defense repertoire, broad interest, and relevance to Nature Communications). The authors are grateful to the reviewer for very constructive comments and

suggestions, which are of great help to the improvement of our manuscript. Below, we provide detailed responses to each of the comments and suggestions.

Comment No.2 Abstract and Title

Title: Consider refining your title so it incorporates both the scale of your sampling and the central novel findings about aquifer prokaryotic defenses. For instance, highlight that you conducted a large or nationwide survey, underscore the discovery of previously unknown elements in the defensome, and make clear this work pertains to aquifer ecosystems.

Abstract: It effectively summarizes the goals, methods, and highlights (novel defense genes, phage anti-defense). One improvement is to add a brief concluding sentence on biotechnological or ecological implications of these findings (for example, CRISPR-Cas9 applications). Remove from the abstract the phrase “Compared to the NCBI RefSeq complete prokaryotic genome database” (page 2, lines ~26–27) so that the abstract can focus more on the broader significance rather than on specific comparisons.

Response: Thank you very much. Following the reviewer’s suggestion, we highlighted the nationwide sampling and the discovery of previously unknown elements. At the review stage, we stressed these issues in the “Abstract” but temporally maintained the original title, focusing on the unknown prokaryotic defensome in the underexplored groundwater ecosystems. According to the reviewer’s suggestion, we modified the Abstract to include a brief concluding sentence and removed the phrase “Compared to the NCBI RefSeq complete prokaryotic genome database”. The revised Abstract reads:

“Groundwater harbors a pristine biosphere where microbes co-evolve with less human interference, yet the ancient and ongoing arms race between prokaryotes and viruses remains largely unknown in such ecosystems. Based on our recent nationwide groundwater monitoring campaign, we construct the first metagenomic groundwater prokaryotic defensome catalogue (GPDC), encompassing 190,810 defense genes,

90,824 defense systems, 139 defense families, and 669 defense islands from 141 prokaryotic phyla. Over 94% of the defense genes in GPDC are novel and contribute vast microbial immune resources in groundwater. We find that candidate phyla radiation (CPR) bacteria possess higher defense system density and diversity against intense phage infection, while microbes as a whole exhibit an inverse relationship between defense systems and adaptive traits like resistance genes in groundwater. We further identify five first-line defense families covering 69.2% of the total defense systems, and high-turnover accessory immune genes are mostly conveyed to defense islands by mobile genetic elements. Our study also reveals viral resistance to microbial defense through co-localized anti-defense genes and interactions between CRISPR-Cas9 and anti-CRISPR protein. These findings expand our understanding of microbial immunity in pristine ecosystems and offer valuable immune resources for potential biotechnological applications.”

Comment No.3 Introduction and Background

The introduction provides context for phage–microbe coevolution, but some points need refinement. The authors should avoid labeling CPR and DPANN as “keystones” without formal analyses. According to a robust body of literature (Banerjee et al., 2018; Herren & McMahon, 2018; Röttjers & Faust, 2018; and others), identifying keystone taxa requires network analysis, machine learning, functional omics, or experimental validation. If the manuscript retains the term “keystone,” the authors should justify it with evidence or remove the term to avoid misinterpretation.

Have caution when describing these groups as symbionts or widely distributed if the references are not conclusive.

Page 4, line 78: the authors’ statement that CPR and DPANN are symbionts is very strong, so they should verify whether multiple lines of evidence support this lifestyle.

Page 4, line 80: once again, great caution is advised regarding the term “keystone,” and it should be clearly demonstrated if claimed.

Page 4, line 88: they are labeled “abundant and widespread symbionts,” but the manuscript should include multiple references if making such a general assertion.

Readers outside specialized fields might need more detail on why CPR and DPANN are important and how frequently they appear in subsurface environments. Adding more citations on aquifer microbial ecology (Page 2, Lines ~28–33) can strengthen the introduction, and clarifying the role of these lineages at Page 2, ~Line 40 would help readers understand their significance.

Response: Thank you very much. To avoid confusion, we removed the term “keystone” in the revised manuscript. Recently, CPR and DPANN received extensive attention as “keystone taxa”¹⁰ and typical symbionts particularly in subsurface environments. In the revised version, we provided additional references on their symbiotic lifestyle, preference in groundwater habitats, and importance in understanding natural microbiome function (Lines 73-81), which reads:

“Candidate phyla radiation (CPR) bacteria and DPANN (an acronym formed from the initials of the first five lineages discovered: Diapherotrites, Parvarchaeota, Aenigmarchaeota, Nanohaloarchaeota, and Nanoarchaeota) archaea represent ultrasmall symbiotic microorganisms^{31–33} mostly detected in oxygen-limited environments, particularly in groundwater ecosystems^{34,35}. Genome analyses^{35,36} and experimental studies^{31,32} reveal their distinctive features: reduced genome sizes and notable gaps in core metabolic potential, consistent with their symbiotic lifestyle. Understanding these microorganisms is important because their interactions with other microorganisms are likely to shape natural microbiome functions³².”

Meanwhile, more references on aquifer microbial ecology are added to highlight the uniqueness of this environment (Lines 69-70), which reads:

“Groundwater is a critical freshwater resource, with microbes as the dominant living organisms^{25,26} playing a crucial role in global biogeochemical cycling processes^{27,28}.”

Comment No.4 Methods

Another major concern is the 0.22 μm filter size. Some prior studies used 0.1 μm filters to capture ultra-small prokaryotes (such as CPR). The difference in filter pore size might exclude some smaller cells, and the authors should acknowledge any potential underestimation of CPR/DPANN lineages. This point is highlighted, for instance, by Luef et al. (2015) regarding ultra-small bacterial cells in groundwater.

The sampling plan (607 groundwater samples, 525 new wells) is substantial, covering multiple geological zones, thus supporting large-scale claims. Metagenomic methods (metaWRAP, checkM, dRep, DefenseFinder) are appropriately described, although the authors should justify thresholds such as $\geq 70\%$ completeness and $\leq 10\%$ contamination more explicitly, and discuss the possibility of incomplete recovery of multi-gene defense systems in partially assembled genomes.

Please clarify abbreviations like $QS > 70$ (on page 19) and specify precisely what the authors tested via stepwise linear regression (page 22, lines 509–510). In addition, the authors mentioned they conducted all analyses in R (v4.3.1) without providing detailed scripts. Making the R code or an equivalent workflow public would be important for reproducibility.

Response: Thank you very much. We acknowledge that the use of 0.22 μm filter size may limit the capture of ultra-small prokaryotes. However, this choice reflects a necessary trade-off between microorganism recovery and practical feasibility (huge resistance during filter). Since we aim to broadly characterize microbial defense systems across groundwater communities, rather than exclusively targeting ultra-small taxa, the use of 0.22 μm filter size is acceptable for the purpose of this study with a broader scope and very large filtered volume (2000 L). Even in the studies^{11–13} focusing

on viruses which are smaller than almost all prokaryotes, 0.22 μm membrane is commonly used and accepted due to similar technical constraints. Using a 0.1 μm filter would dramatically increase filtration resistance, making it difficult to process sufficient water volumes to obtain enough biomass for DNA extraction and robust metagenomic analyses. In this case, such resistance would impact the success of sequencing library construction and subsequent analyses. Additionally, the formation of filter cake on the membrane during filtration helps retain smaller cells, partially mitigating the pore-size limitation. To acknowledge the limitation of 0.22 μm filter size, we revised the relevant part in Methods section (Lines 427-429):

“Given the exceptionally large volume (2000 L) of water filtered and our focus on broad microbial communities, the potential loss of some ultra-small cells (CPR/DPANN) due to use of a 0.22 μm filter is considered an acceptable trade-off.”

Thank you for your appreciation of our sampling efforts. Regarding genome quality thresholds ($\geq 70\%$ completeness and $\leq 10\%$ contamination), we provided further justification, primarily on the multigenic nature of defense systems, which require more stringent completeness standards than general genome analyses (typically 50% completeness) (Lines 458-460):

“Given that defense systems can be multigenic, a stricter completeness threshold ($\geq 70\%$) was employed in this study compared to the conventional 50% threshold used in other metagenomic analyses^{98,99}.”

We acknowledge that partially assembled genomes may lead to the incomplete recovery of multi-gene defense systems. This limitation has been discussed in the manuscript (Lines 460-462):

“Despite applying a higher completeness threshold, partially assembled genomes may still result in the incomplete recovery of multi-gene defense systems and certain defense islands.”

We have clarified all abbreviations and specified the variables tested in the stepwise linear regression (Lines 568-571):

“Stepwise linear regression analyses were performed to examine the relationship between the number of defense genes/systems, genome size, phylogenetic depth, and N50 using the *lm* and *step* functions in the *stats* package.”

To ensure reproducibility, all Shell scripts and R codes used in this study are publicly available on GitHub (<https://github.com/lianmsu/aquifer-defensome>).

Comment No.5 Results

Vast and Novel Defensome: Approximately 94% of detected defense genes are novel, which indicates a large expansion of microbial immune diversity in groundwater. The supplementary figures (S1–S3) show distributions and correlations with genome size.

Lineage-Specific Defense Patterns: CPR and DPANN present high densities of defense systems, but the authors must be cautious about labeling them as “keystone” if they have not used recognized methods (network analysis, etc.). Large-genome “defense supercarriers” (e.g., certain Pseudomonadota) mirror other environments but also hint at unique aquifer conditions.

Trade-Off with Adaptive Traits: The inverse correlation between defense systems and ARGs, MRGs, or virulence factors is a key ecological finding, pointing toward possible CRISPR-based inhibition of plasmid-borne traits.

Phage Anti-Defense Genes: Observations of anti-CRISPR, anti-RM, anti-Thoeris underlie a complex virus–host arms race in aquifers. Figures S4–S5 (protein structures, Cas9 alignment) reinforce the novelty of these potential interactions.

It was suggested to expand the discussion on whether large genome size correlates with frequent phage encounters (Page 7, Lines ~145–150). Also, consider clarifying the biological significance of functional annotations like COG L, K, and V (lines 281–282).

Partial MAGs might underestimate certain defense islands, so the authors could comment on that limitation as well.

Response: Thank you very much. As we have responded to the preceding questions, the term “keystone” was removed in the revised manuscript. As suggested, we conducted further analyses on the relationship between genome size and phage encounters, revealing a significant positive correlation. This aligns with prior evidence¹⁴ that larger genomes usually undergo more horizontal gene transfer (HGT) events, including phage encounters. Tesson and co-authors⁸ have also found a significant positive correlation ($P < 0.0001$) between genome size and prophage number in NCBI RefSeq genomes, reinforcing these findings. In the revised version, we expanded the discussion as follows (Lines 162-166):

“Our results (Fig. S7A) also show a significant positive correlation ($\rho = 0.88$, $P = 0.0007$) between genome size and prophage encounters. These patterns likely reflect the greater need for defense systems in bacteria with larger genomes, which tend to experience higher levels of invasion by exogenous genetic materials^{47–49}.”

Fig. S7. Effects of genome size on prophage encounter and prophage number on the defensome. (A) Variation in prophage number (per MAG) across all MAGs. Error bars represent standard deviations of the mean; correlation assessed by two-sided Spearman's rank test. To ease visualization, standard errors were divided by 10, and the orange points were excluded from the Spearman test. (B) Variation in prophage number (per MAG) across all bacterial phyla. Each dot represents a phylum. (C) Relationship between virus-host abundance ratios (VHR) and MAG size. (D), (E), (F) Positive correlations between defense system (DS) number, DS density, defense family number, and prophage number. Shaded areas represent the 95% confidence interval.

To enhance understanding, we have clarified the functional significance of COG categories L, K, and V, respectively (Lines 348-354):

“Genes annotated as L, K, and V play important roles in cellular information storage and processing, cellular processes and signaling⁷⁶. Their prevalence (Fig. S16A) suggests a functional link to microbial immunity, potentially representing unidentified defense genes. KEGG and PFAM annotations (Fig. S16B-C) further highlight defense-related domains, such as type I restriction-modification DNA specificity domain (K01154), ATP-dependent DNA helicase (K03655), and WYL transcriptional regulators^{77,78}, underscoring their connection to microbial defense²².”

We also indicated the limitation that partially assembled MAGs may underestimate certain defense islands and multi-gene defense systems, with corresponding changes in the revised version (Lines 460-462):

“Despite applying a higher completeness threshold, partially assembled genomes may still result in the incomplete recovery of multi-gene defense systems and certain defense islands.”

Comment No.6 Discussion and Conclusion

The discussion links intense viral predation with observed patterns of defense density and describes a framework of “cornerstone” vs. “accessory” defense families. Caution is necessary regarding the term “keystone,” since it has a precise meaning in microbial

ecology. If Results and Discussion are merged, the text must still maintain a clear logical flow.

I suggest merging the current Results and Discussion into a single combined section, as the Results already contain interpretive material and references. This reorganization could help the manuscript read more fluidly and concisely. Where partial MAGs are concerned, the authors should note how multi-gene defenses such as CBASS might be difficult to detect in fragmented assemblies.

Finally, the conclusion highlights potential biotechnology applications (e.g., novel Cas9 variants) and the relevance for managing antibiotic resistance (via phage–ARG interactions).

Response: Thank you very much for your valuable comments. We removed the term “keystone” throughout the revised manuscript. As suggested, we merged the Results and Discussion sections to improve readability and maintain a logical flow. We agree that incomplete genomes may limit the detection of multi-gene defense systems or defense islands, and we indicated this limitation in the manuscript (Lines 460-462):

“Despite applying a higher completeness threshold, partially assembled genomes may still result in the incomplete recovery of multi-gene defense systems and certain defense islands.”

Comment No.7 Language, Structure, and Formatting

I found minor grammatical or typographical issues (e.g., line 30, line 346), so a thorough proofreading is advised.

Figures in the main text can be visually dense; minor modifications (like clarifying labels or using consistent color scales) will improve accessibility.

In the supplementary figures, explicitly linking them to points raised in the text (for instance, “see Fig. S1–S3 for extended distribution analyses”) would help readers navigate the data.

Response: Thank you very much. We have carefully proofread the manuscript to correct grammatical and typographical issues. Figures have been refined for improving clarity and consistency (notably in **Figure 5**). Following your suggestion, we have explicitly linked supplementary figures to relevant sections in the text to improve accessibility.

Comment No.8 Specific Comments Based on Pages and Line Numbers

Page 2, Lines ~28–33: Insert references on aquifer microbial ecology to highlight the uniqueness of this environment.

Page 5, Lines ~90–93: Clearly define your threshold for labeling a defense gene “novel.” For example, is it <80% identity and coverage when compared to RefSeq?

Page 7, Lines ~145–150: Note whether “supercarrier” genomes share certain ecological traits, such as higher metabolic versatility or niche specialization that increases the need for diverse antiviral defenses.

Page 9, Lines ~205–210: Briefly discuss the mechanistic link between CRISPR-based defenses and plasmid-borne ARG acquisition. If the authors have plasmid data in the supplementary material, direct readers there.

Supplementary Figures S4–S5: Including AlphaFold confidence metrics (e.g., pLDDT scores) for protein structure predictions would enhance confidence in the conclusions about novel anti-defense or Cas proteins.

Response: Thank you very much. Following your suggestions, we have incorporated additional references on aquifer microbial ecology to highlight the uniqueness of this environment (**Lines 69-70**), which reads:

“Groundwater is a critical freshwater resource, with microbes as the dominant living organisms^{25,26} playing a crucial role in global biogeochemical cycling processes^{27,28}.”

As far as the thresholds, we initially used the threshold of <80% coverage and <90% identity compared to RefSeq, considering these parameters are standard in protein clustering analyses. Furthermore, we tested a more stringent criterion of <80% coverage and <80% identity, yielding 84.2% sequence novel genes in bacteria and 93.7% in archaea, without altering the overall conclusions. To ensure clarity, we have revised the relevant part in the Methods section to explicitly define “novelty” (Lines 481-483) as follows:

“The blastp results were then filtered using <80% coverage and <90% identity to identify sequence novelty of groundwater defense genes, considering these parameters are commonly employed in protein clustering analyses.”

We found defense supercarriers exhibit larger genome sizes, longer average gene lengths, and a higher number of coding sequences compared to other bacteria. Prior research¹⁴ suggests that larger genomes experience more HGT events, which may explain their accumulation of defense genes. Additionally, we annotated their optimal growth temperature and found that supercarriers tend to have slightly lower values, indicating distinct physiological traits. These characteristics are discussed in the revised version (Lines 131-135 and 166-167, respectively), which reads:

“These “defense supercarriers” are mainly annotated as Pseudomonadota, possessing larger genome size, longer average gene lengths, and more coding sequences than other bacteria (Fig. S5A-D). Moreover, they exhibit distinct physiological traits like lower optimal growth temperatures (Fig. S5E).”

“In this regard, the larger genome size of “defense supercarriers” implies their increased need for defense.”

Fig. S5. Bacterial supercarriers of defense systems. (A) Taxonomic composition of the 268 supercarrier genomes. **(B), (C), (D) and (E)** Comparison of genome size (Mb), total coding sequences, average gene length (bp), and optimal growth temperature (°C) between supercarriers and other bacterial genomes. Significant differences were assessed using the Wilcox test with Bonferroni-adjusted P values. “****” indicates $P < 0.0001$.

Following the reviewer’s suggestion, we strengthened the discussion regarding the possible inhibition of MGE-carried ARGs by defense systems in the revised version (Lines 285-290), which reads:

“On one hand, these immune systems (e.g., CRISPR) protect bacteria and archaea against viruses and other mobile genetic elements (MGEs)^{24,65}, thus curbing the acquisition of adaptive traits carried by other MGEs, such as plasmid-borne ARGs^{66–68}. On the other hand, bacteria with limited defense are likely to be infected by these genetic elements, which allows frequent HGT of resistance genes and virulence factors from genetic elements to bacterial genomes.”

As suggested, we also included the AlphaFold3 confidence metrics for protein structure predictions, enhancing the confidence in the conclusions about these anti-defense genes. The predicted template modeling (pTM) score and pLDDT score was shown in the revised version (Fig. S19).

Comment No.9 Specific Suggestions for Improvement

Explain filter size bias (0.22 μm vs. 0.1 μm) and any underestimation of ultra-small prokaryotes. Use recognized methods or references if calling CPR and DPANN “keystone”; if these methods are not available, remove that term to avoid confusion. Discuss partial MAG assemblies in the context of multi-gene defense systems. This helps readers interpret possible underestimation of complex systems like CBASS. Merge or reorganize the Results and Discussion if you desire a more streamlined structure, as the current Results section contains numerous interpretive statements. Ensure reproducibility by providing R code or an equivalent workflow in a public repository, clarifying the statistical methods (for example, your stepwise linear regression) and data processing steps. Consider summarizing “cornerstone vs. accessory” defense families in a concise table that describes abundance, possible functions, and typical genomic locations.

Response: Thank you very much for your valuable and specific suggestions. As explained in the point-by-point responses above, we used a 0.22 μm filter to facilitate processing large water volumes while minimizing downstream analyses challenges. We have also included a discussion about the potential underestimation of ultra-small prokaryotes in Lines 427-429. To avoid misinterpretation, we removed the term “keystone” in reference to CPR/DPANN. Discussions are also given about the potential effects of partial metagenomic assembly on the recovery of multi-gene defense systems and defense islands in Lines 458-462. In the revised manuscript, we have merged Results and Discussion to streamline the manuscript. All wrapper Shell scripts and R codes used in this study have been made publicly available on GitHub

(<https://github.com/lianmsu/aquifer-defensome>). The statistical methods and data processing steps have been clarified to ensure clarity (Lines 568-571). Meanwhile, we summarized the two types of defense systems (first-line vs. accessory) and their associated metadata in Supplementary Data 9.

Comment No.10 Overall Recommendation

Recommendation: Major Revisions

Explanation: The study is data-rich and methodologically thorough, presenting a novel view of aquifer microbial immunity. Nonetheless, important clarifications are needed before publication in Nature Communications: (1) Remove or justify the use of “keystone” lineages with formal analyses (e.g., network-based or machine-learning-based identification). (2) Address possible biases in filter size (0.22 μm) and partial MAGs, which may exclude key taxa or omit multi-gene defenses. (3) Improve clarity and structure, for example by merging the Results and Discussion sections if that would reduce redundancy. (4) Strengthen reproducibility by providing code and clarifying all thresholds (especially for “novelty” definitions and statistical methods).

With these revisions in place, the manuscript will provide a significant contribution to our understanding of viral-host interactions in subsurface ecosystems and will be well-suited for publication in Nature Communications.

Response: Again, the authors are very grateful to the Reviewer #3. Following the reviewer’s very constructive comments and suggestions, important clarifications have been made in the revised manuscript, as shown in the above point-by-point responses. We hope the modified manuscript can meet the requirements of the journal *Nature Communications*.

References

1. DeWeirdt, P. C., Mahoney, E. M. & Laub, M. T. DefensePredictor: A machine learning model to discover novel prokaryotic immune systems. 2025.01.08.631726 Preprint at <https://doi.org/10.1101/2025.01.08.631726> (2025).
2. Beavogui, A. *et al.* The defensome of complex bacterial communities. *Nat. Commun.* **15**, 2146 (2024).
3. Cury, J. *et al.* Conservation of antiviral systems across domains of life reveals immune genes in humans. *Cell Host Microbe* **32**, 1594-1607.e5 (2024).
4. Dai, R. *et al.* Crop root bacterial and viral genomes reveal unexplored species and microbiome patterns. *Cell* **0**, S0092867425001989 (2025).
5. Mutte, S. K., Barendse, P., Ugarte, P. B. & Swarts, D. C. Distribution of bacterial DNA repair proteins and their co-occurrence with immune systems. *Cell Rep.* **44**, 115110 (2025).
6. Getz, L. J., Fairburn, S. R., Vivian Liu, Y., Qian, A. L. & Maxwell, K. L. Integrins are anti-phage defence libraries in *Vibrio parahaemolyticus*. *Nat Microbiol* 1–10 (2025) doi:10.1038/s41564-025-01927-7.
7. Gaborieau, B. *et al.* Prediction of strain level phage–host interactions across the *Escherichia* genus using only genomic information. *Nat. Microbiol.* **9**, 2847–2861 (2024).
8. Tesson, F. *et al.* Systematic and quantitative view of the antiviral arsenal of prokaryotes. *Nat. Commun.* **13**, 2561 (2022).
9. Zhang, Q. *et al.* Long-term metagenomic insights into the roles of antiviral defense systems in stabilizing activated sludge bacterial communities. *ISME J.* wraf051 (2025) doi:10.1093/ismejo/wraf051.
10. Zhong, S. *et al.* May microbial ecological baseline exist in continental groundwater? *Microbiome* **11**, 152 (2023).
11. An, L. *et al.* Global diversity and ecological functions of viruses inhabiting oil reservoirs. *Nat. Commun.* **15**, 6789 (2024).
12. Wu, Z. *et al.* Unveiling the unknown viral world in groundwater. *Nat. Commun.* **15**, 6788 (2024).
13. Hwang, Y., Roux, S., Coclet, C., Krause, S. J. E. & Girguis, P. R. Viruses interact with hosts that span distantly related microbial domains in dense hydrothermal mats. *Nat. Microbiol.* **8**, 946–957 (2023).
14. Oliveira, P. H., Touchon, M. & Rocha, E. P. C. Regulation of genetic flux between bacteria by restriction–modification systems. *Proc. Natl. Acad. Sci.* **113**, 5658–5663 (2016).

Response to the Reviewers' comments

Response to the Reviewer #1

The authors addressed all my questions/suggestions.

Congratulations for the work.

Response: We sincerely thank the reviewer #1 for the positive comments and encouraging remarks. We greatly appreciate the time and effort dedicated by the reviewer with excellent suggestions which are indeed valuable in improving the quality and clarity of our manuscript.

Response to the Reviewer #2

The authors have addressed most of my concerns in their revised manuscript and they have made substantial improvements to their study. I was especially pleased to see the efforts to experimentally validate several defense systems via recombinant phage infection (plaque) assays in *E. coli*. However, a remaining weakness of the study is the lack of parallel omics data to show which systems are actively deployed in situ (this was not directly addressed in the authors' response). I imagine such datasets as metaproteomes or metatranscriptomes may not be available from the original sampling effort, and as such, couldn't be provided for a revision of this study.

Response: Thank you very much for acknowledging our revisions and improvements on the manuscript. We sincerely appreciate your positive feedback, particularly regarding our experimental validation of defense systems using plaque assays in *E. coli*. We fully agree that the inclusion of parallel omics data would enhance our understanding of the functional dynamics of defense systems in the environment. We will make it a priority to integrate such approaches in future studies. We are grateful to Reviewer #2 for the insightful suggestions, which have significantly contributed to the improvement of this work.

Response to the Reviewer #3

Thank you for the thorough revision. You have addressed all substantive issues raised in the first round. In particular, the “keystone” terminology has been removed or appropriately qualified; the potential 0.22 μm -filter bias and limits of partial MAG assemblies are now discussed; sequence-novelty thresholds, AlphaFold confidence metrics and every software parameter are clearly stated; the full R/Shell workflow is openly available; and you added plaque-assay validation for three representative defence systems (R-M IIG, SoFIC and CBASS), which substantiates the in-silico predictions. The Results and Discussion sections have been merged, improving flow and eliminating redundancy, and language and figure legends have been polished.

Only very small editorial items remain (for example, add units to the y-axis of Supplementary Fig. S6 panel D; provide a short Materials-and-Methods subsection specifically for the plaque assays; insert hyphens in “R-M IIG” where needed). These can be handled easily during production.

With those minor tweaks, the manuscript is ready for publication in Nature Communications.

Response: Thank you very much for your thorough review and thoughtful summary of our revision. We sincerely appreciate your constructive comments and suggestions, which have been instrumental in significantly improving the quality and clarity of this work. In response to the remaining editorial points you pointed, we have made the following adjustments in the revised version:

- Units have been added to the y-axis of Supplementary Fig. S6 panel D;
- A dedicated subsection (i.e., Experimental validation of defense systems) has been included in the Methods to describe the plaque assay procedures;
- Hyphens have been consistently added to “R-M IIG” as appropriate.

We hope these final revisions meet the editorial standards of *Nature Communications*.
Again, we would like to thank the reviewers for their valuable contributions throughout
the review process.

P
A
G
E

\
*